# Health status and psychological outcomes after trauma: A prospective multicenter cohort study

Nena Kruithof[1]*, Suzanne Polinder[2], Leonie de Munter[1], Cornelis L. P. van de Ree[1], Koen W. W. Lansink[1,3,4], Mariska A. C. de Jongh[1,4], BIOS-group[¶]

1 Department Trauma TopCare, ETZ Hospital (Elisabeth-TweeSteden Ziekenhuis), Tilburg, the Netherlands,
2 Department of Public Health, Erasmus University Medical Centre, Rotterdam, the Netherlands,
3 Department of Surgery, ETZ Hospital (Elisabeth-TweeSteden Ziekenhuis), Tilburg, the Netherlands,
4 Brabant Trauma Registry, Network Emergency Care Brabant, Tilburg, the Netherlands

¶ Membership of the BIOS group is provided in the Acknowledgements.
* nenakruithof@hotmail.com

**Data Availability Statement:** Data of the BIOS are unsuitable for public deposition due to the privacy of participant data. Data are anonymized, but due to relatively few severe cases, patients could be

## Abstract

### Introduction

Survival after trauma has considerably improved. This warrants research on non-fatal outcome. We aimed to identify characteristics associated with both short and long-term health status (HS) after trauma and to describe the recovery patterns of HS and psychological outcomes during 24 months of follow-up.

### Methods

Hospitalized patients with all types of injuries were included. Data were collected at 1 week 1, 3, 6, 12, and 24 months post-trauma. HS was assessed with the EuroQol-5D-3L (EQ-5D-3L) and the Health Utilities Index Mark 2 and 3 (HUI2/3). For the screening of symptoms of post-traumatic stress, anxiety and depression, the Impact of Event Scale (IES) and the Hospital Anxiety and Depression Scale (HADS) subscale anxiety (HADSA) and subscale depression (HADSD) were used. Recovery patterns of HS and psychological outcomes were examined with linear mixed model analyses.

### Results

A total of 4,883 patients participated (median age 68 (Interquartile range 53–80); 50% response rate). The mean (Standard Deviation (SD)) pre-injury EQ-5D-3L score was 0.85 (0.23). One week post-trauma, mean (SD) EQ-5D-3L, HUI2 and HUI3 scores were 0.49 (0.32), 0.61 (0.22) and 0.38 (0.31), respectively. These scores significantly improved to 0.77 (0.26), 0.77 (0.21) and 0.62 (0.35), respectively, at 24 months. Most recovery occurred up until 3 months. At long-term follow-up, patients of higher age, with comorbidities, longer hospital stay, lower extremity fracture and spine injury showed lower HS. The mean (SD) scores of the IES, HADSA and HADSD were respectively 14.80 (15.80), 4.92 (3.98) and 5.00

identified. Therefore, BIOS data are available for any interested researcher who meets the criteria for access to confidential data. The Brabant Trauma Registry (e-mail: secretariaat@nazb.nl) may be contacted to request data.

**Funding:** This work was supported by The Netherlands Organization for Health Research and Development (ZonMw) under grant number 80-84200-98-14225. https://www.zonmw.nl/nl/ The funders had no role in study design, data collection and analysis, decision to publish, or preparation of the manuscript.

**Competing interests:** The authors have declared that no competing interests exist.

(4.28), respectively, at 1 week post-trauma and slightly improved over 24 months post-trauma to 10.35 (14.72), 4.31 (3.76) and 3.62 (3.87), respectively.

## Discussion

HS and psychological symptoms improved over time and most improvements occurred within 3 months post-trauma. The effects of severity and type of injury faded out over time. Patients frequently reported symptoms of post-traumatic stress.

## Trial registration

ClinicalTrials.gov identifier: NCT02508675.

## Introduction

Trauma poses a large burden on public health [1]. Reduction of trauma-related mortality in high-income countries [2] has resulted in increased numbers of trauma survivors with long-term injury impact [3], including reduced health status (HS) [4]. An improved understanding of the quality of survival of patients is critically important for improving health care quality and in evaluating trauma care. Furthermore, it is important to understand short and long-term recovery patterns of HS in terms of injured patient characteristics and to identify predictors of outcome of seriously injured patients [5, 6].

Establishing recovery patterns in the short and long-term requires longitudinal data [7]. Non-fatal outcomes after trauma can be assessed with an overall measure of HS. HS includes patients' physical functioning, state of mind and social activities [8]. In general, trauma has a large impact on HS [4, 9–12], but large variations between patients have been observed [11, 12].

In addition, several relevant non-fatal outcomes after trauma may be assessed more specifically. These include psychological outcomes, such as anxiety and depression. Psychological problems are often reported among trauma patients [13–18] and are associated with worse HS [9, 16].

The fact that trauma has an impact on diverse aspects of patient health, illustrates that a multidimensional approach is necessary for a comprehensive understanding of non-fatal outcomes after trauma. This also allows studying the mutual relations between non-fatal outcomes. Using a multidimensional approach to measure outcomes including HS and symptoms of depression, anxiety and post-traumatic stress will result in a comprehensive understanding of non-fatal outcomes after trauma. In addition, to assess prognostic factors for a poor outcome it is important to cover the entire spectrum of the trauma population without exclusion of particular patient groups (e.g. elderly). The number of longitudinal cohort studies that examine multiple non-fatal outcomes in a large sample with a broad inclusion of type and severity of injury is limited [12, 19–23]. Most studies start measuring outcomes at least 3 months after trauma, resulting in little knowledge about the real short-term consequences [11, 24–27].

The overall aim of the Brabant Injury Outcome Surveillance (BIOS), a population based longitudinal study, is to provide more insight into recovery patterns and determinants of non-fatal outcomes after trauma. The aims of this population-based study are 1. to identify characteristics associated with short, mid and long-term HS and 2. to describe the 2 year recovery

patterns of HS and psychological outcome for different categories of trauma patients. This information is important for understanding the short and long-term recovery patterns and for best informing provision of trauma care to injured patients with long-term disability.

## Methods

### Study design and participants

Data were obtained from the BIOS. The BIOS-study is a prospective observational cohort study in which HS and psychological outcomes are assessed in injured patients in the first 24 months after trauma. The methods of the BIOS have been described in detail in a published research protocol (doi: 10.1136/injuryprev-2016-042032) [28].

Recruitment occurred in all ten hospitals of the Noord-Brabant region (the Netherlands) from August 2015 until November 2016. Adults (≥18 years) who visited an emergency department ≤48 hours after trauma and who were admitted to the hospital, were invited to participate. All types of injuries were included regardless of the intent or severity of the injury. Patients who died between hospital discharge and the first week post-trauma, non-Dutch speaking patients, patients with no permanent address or patients with a pathological fracture were excluded. A proxy informant (caregiver or family member) was asked to complete the self-administered questionnaires if the patient was incapable of participating in the BIOS study him- or herself. Proxy informants were invited to enroll in the study 1 month post-trauma. Informal caregivers (e.g. family members) and paid caregivers (e.g. nurses) were allowed to function as proxy informants.

The Brabant Trauma Registry (BTR) compiles pre-hospital and hospital data of all trauma patients admitted after presentation to the ED in the Noord-Brabant region. Quality of the data of the BTR and BIOS was checked on outliers and completeness by a trauma coordinator and researcher respectively. Data of the BTR was checked as part of routine practice while the data of the BIOS was checked randomly by the researchers. Furthermore, data from a sample of the trauma registry was checked manually by a trauma surgeon.

The study was approved by the Medical Ethics Committee Brabant, the Netherlands (project numbers NL50258.028.14 and NW2016-09). Prior to participation, participants signed an informed consent form.

### Data collection

Questionnaires were sent at 1 week and 1, 3, 6, 12 and 24 months after trauma. Based on the participants' preference, follow-up questionnaires were either completed by paper and pencil or digitally. The questionnaires collected data on general patient characteristics (date of birth, gender), self-reported comorbidities (by using a modified version of the Cumulative Illness Rating Scale [29]), self-reported HS (i.e. EuroQol-5D-3L (EQ-5D-3L) [30], the Health Utilities Index (HUI) Mark 2 and Mark 3 [31]) and self-reported psychological functioning (i.e., Hospital Anxiety and Depression Scale (HADS) [32] and the Impact of Event Scale (IES) [33]). Proxy informants did not complete questionnaires regarding psychological outcome.

To increase the response rate, patients who did not complete a questionnaire up until 3 or 6 months post-trauma were asked to complete a short version of the BIOS-questionnaire. Patients who completed the shortened questionnaire included those who could not be reached by phone and did not return a BIOS questionnaire. In this short questionnaire, educational level, comorbidities, the EQ-5D-3L and the IES were included.This short questionnaire did not include proxy assessments. In the shortened questionnaire, pre-injury HS was not collected.

If participants did not complete the questionnaire, they were not excluded from the study but they were still invited at the subsequent time points.

## Outcome measures

The EQ-5D and HUI are used in various studies measuring HS after trauma [9, 11, 24, 26, 34–39]. The EQ-5D provides valid results for trauma patients when it is completed by a proxy informant [40]. Moreover, a combination of the EQ-5D and the HUI is recommended for use in trauma patients since the combination of these measures covers all relevant dimensions of health [37, 41].

The EQ-5D consists of the EQ-5D descriptive system and the EQ-visual analogue scale (EQ-VAS). The EQ-5D comprises the following five dimensions: 'mobility', 'self-care', 'usual activities', 'pain/discomfort' and 'anxiety/depression'. Each dimension can be scored as 'no problems', 'moderate problems' or 'severe problems' [30]. A scoring algorithm is available by which each HS description can be expressed as a summary score. This summary score ranges from 0 for death and 1 for full health and can be interpreted as a judgment on the relative desirability of an HS compared with perfect health. A summary score of these five dimensions (EQ-5D utility) can be calculated by using the Dutch tariffs [42]. The EQ-VAS is a vertical visual analogue scale with 0 indicating the worst imaginable health state and 100 indicating the best imaginable health state. The EQ-5D and EQ-VAS were also measured pre-injury, by asking participants 1 week or 1 month and proxy informants 1 month after the trauma for the patients' HS before sustaining the injury. The EQ-VAS was not included in the short questionnaire.

The HUI is a self-administered HS questionnaire that covers the main health domains that are affected by injury, with a particular focus on functional capacities. The HUI consists of 15 questions, classifying respondents into either the HUI2 or HUI3 health states [31]. Single-attribute and overall HS utility scores are calculated using the respective HUI2 and HUI3 utility functions. The results of the HUI questionnaires are converted by an algorithm into the levels of the complementary HUI2 and HUI3 classification system to form seven-element and eight-element health state vectors. From these vectors, single-attribute and overall health state utility scores are calculated [31].

For both the EQ-5D and the HUI, a scoring algorithm is used in which a score of 1 represents full health, 0 represents death and negative values indicate a HS of worse than death [30, 31].

The Hospital Anxiety and Depression Scale (HADS) was used to assess symptoms of anxiety and depression [32]. The HADS consists of 14 questions, 7 for symptoms of anxiety (HADSA) and 7 for depressive symptoms (HADSD). All questions have a 4-point response scale (0–3) and the scores for both subscales ranged from 0 to 21. A higher subscale score indicates greater severity of symptoms for anxiety and depression with a subscale value of $\geq 11$ indicating a probable case (i.e., clinical symptoms) [32].

The IES was used to assess self-reported symptoms of post-traumatic stress [33]. The IES consists of 15 items of which the patient could use a 4-point scale (0 = not at all, 1 = rarely, 3 = sometimes and 5 = often) to determine whether the statement is present during the last seven days. The IES measures intrusive re-experience of the trauma and avoidance of trauma-related stimuli. A total sum score for the IES could be calculated ranging from 0 to 75. A score of $\geq 35$ is considered as having symptoms of post-traumatic stress [33].

## Prognostic factors

Hospital length of stay (LOS), admission to an Intensive Care Unit (ICU) and type and severity of injury were collected from the Brabant Trauma Registry and merged with the BIOS-data.

The Abbreviated Injury Scale (AIS) codes (AIS-90, update 2008) [43] were used to create 14 injury group classifications (e.g. hip fracture, severe abdominal injury) representing the most common types of injuries (see S1 Table). Patients who suffer multiple injuries could be classified into one or more injury group classifications.

Trauma severity was based on the Injury Severity Score (ISS). The ISS is based on the square of the highest Abbreviated Injury Scales (AIS) scores of the three most severely injured body regions with a range of 1 to 75. An ISS of $\geq 16$ is considered severely injured [43].

To determine socio-economic status (SES), educational level was used. Educational level was categorized into three levels; low (primary education, preparatory secondary vocational education or without diploma), middle (university preparatory education, senior general secondary education or senior secondary vocational education and training), and high (academic degree or university of applied science).

## Statistical analyses

Patients were included in the analyses if they completed a questionnaire for at least one of the predetermined time points. For the non-responders of the BIOS, we could not obtain educational level. Therefore, in the non-responders analysis, status scores from 2014 were used as a proxy to indicate SES. Status scores were based on the mean income, % of people with a low income, % of people with low educational level and % of unemployed people in the neighborhood. In 2014, the mean status score in the Netherlands was 0.28 [44]. A lower status score indicates a lower SES whereas a higher status score indicates a higher SES. Responders and non-responders were compared on age, gender, status score, ISS, type of trauma, LOS and admission to an ICU using Mann-Whitney U tests and Chi-square tests ($\chi^2$).

Means and standard deviations (SDs) of the EQ-5D-3L, HUI2, HUI3, HADSD, HADSA and IES summary scores were calculated and reported for the total BIOS population and for the different subcategories.

Multiple imputation was conducted with the Multivariate Imputation by Chained Equations procedure [45] to handle missing baseline characteristics and sum scores of the questionnaires due to missing item scores (see S1 File). The dataset was imputed 15 times with 5 iterations. Sensitivity analysis was performed in which only complete cases were included to compare results with the imputed datasets. S2 Table shows the differences between the original and imputed data.

Score options from each dimension of the EQ-5D were dichotomized into 0 = 'no problems' and 1 = 'moderate problems'/'severe problems'.

Four linear mixed models [46] with a random intercepts were performed to assess longitudinal association between prognostic factors and HS over the 24 months after trauma, which were divided into short-term (1 week and 1 month), mid-term (3 and 6 months) and long-term (12 and 24 months) associations. HS was measured with the EQ-5D-3L summary score.

The results were considered statistically significant at a level of $p < 0.05$. All analyses were conducted in SPSS V.24 (Statistical Package for Social Sciences, Chicago, Illinois, USA), except of the multiple imputation which was performed in R version 3.4.0 (The R Project for Statistical Computing).

## Results

### BIOS cohort

During the inclusion period of the BIOS, a total of 10,227 patients were hospitalized because of trauma in one of the participating study centers. Patients were excluded if they did not speak the Dutch language (n = 194), had no permanent address (n = 32), died during their hospital

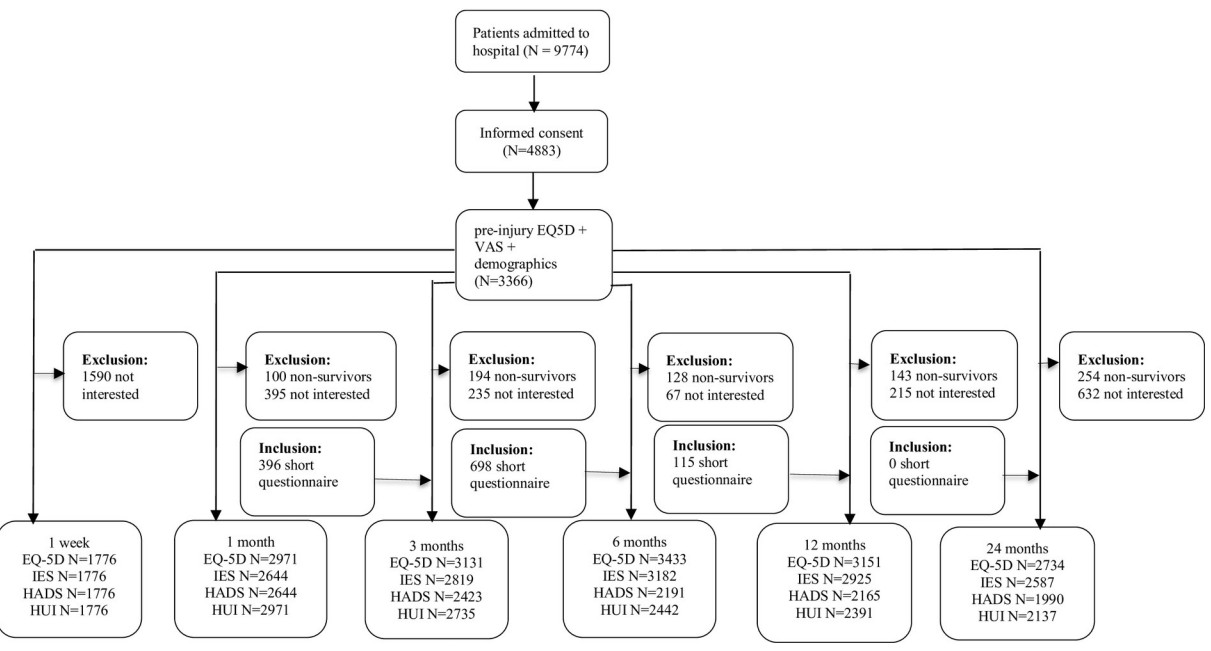

**Fig 1. Number of participants troughout the Brabant Injury Outcome Surveillance (n = 4,883).**

stay within the first week after trauma (n = 219) or had other reasons (n = 8) (e.g., living abroad). Thus, 9,774 patients were eligible for participation in the BIOS, of whom 4,883 patients provided informed consent and were included (50% response rate). Of these 4,883 participants, 1,099 filled out the shortened questionnaires (see Fig 1).

At 1 week and 1, 3, 6, 12 and 24 months following the trauma, we collected data from of 1,776, 2,971, 3,109, 3,418, 3,105 and 2,734 participants (36.4%, 60.8%, 63.7%, 69.9%, 63.6% and 56.0%, respectively, of the study population) (see Fig 1). A total of 1,105 participants (22.6% of the study population) completed all BIOS questionnaires at each time point. In addition, data on pre-injury HS were obtained from 3,366 participants (69% of the study population). After the first week assessment, missing questionnaires were the result of non-response (i.e., patients who had provided no data at any of the previous time points) and loss to follow-up (i.e., patients who had provided data for at least one of the previous time points). The main reason for participants to be lost to follow-up during the study period was that completing the questionnaires was too time consuming. Elderly, participants with low educational levels, longer hospital LOS, moderate injury (ISS 9–15), a hip fracture, severe traumatic brain injury (TBI) and those with severe abdominal trauma showed lower response rates to the 1 week questionnaire but provided data thereafter. In the BIOS, patients aged 18–24 and those who recovered completely were most likely to be lost to follow-up.

## Study population

The median age of the study population was 68 years (IQR 53–80) (Table 1). Responders had a median ISS of 5 (IQR [4–9]) and a large part of the population reported comorbidities. Mild TBI (27.1%) and hip fracture (25.9%) were the most common types of trauma among the participants included in the BIOS. The majority of the participants (n = 2,562, 52.5%) had a low educational level. A total of 407 participants (8% of the study population) were represented by a proxy informant.

**Table 1. Characteristics of responders and non-responders of the Brabant Injury Outcome Surveillance.**

| Characteristics | Responders (n = 4,883) % | Non-responders (n = 4,891) % |
|---|---|---|
| **Gender (male)** | 2,329 (47.7%) | 2,407 (49.0%) |
| **Median age (yrs)** | 68 (IQR 53–80) | 70 (IQR 46–84) |
| 18–24 | 217 (4.4%) | 400 (8.2%) |
| 25–44 | 516 (10.6%) | 767 (15.7%) |
| 45–64 | 1,364 (27.9%) | 1,006 (20.6%) |
| 65–74 | 963 (19.7%) | 563 (11.5%) |
| 75–84 | 1,102 (22.6%) | 1,030 (21.1%) |
| ≥85 | 721 (14.8%) | 1,125 (23.0%) |
| Missing | 0 (0.0%) | 0 (0.0%) |
| **Median SES status score** | 0.33 (IQR -0.24–0.84) | 0.13 (IQR -0.36–0.73) |
| Missing | 60 (1.2%) | 68 (1.4%) |
| **Median days hospital LOS** | 4 (IQR 2–8) | 4 (IQR 2–8) |
| ≤2 | 1,325 (27.1%) | 1,528 (31.2%) |
| 3–7 | 1,944 (39.8%) | 1,642 (33.6%) |
| 8–14 | 937 (19.2%) | 911 (18.6%) |
| ≥15 | 346 (7.1%) | 421 (8.6%) |
| Missing | 331 (6.8%) | 389 (8.0%) |
| **Type of injury** | | |
| Pelvic injury | 293 (6.0%) | 151 (3.1%) |
| Hip fracture | 1,266 (25.9%) | 1,099 (22.5%) |
| Tibia, complex foot or femur fracture | 569 (11.7%) | 505 (10.3%) |
| Shoulder and upper arm injury | 473 (9.7%) | 417 (8.5%) |
| Radius, ulna or hand fracture | 308 (6.3%) | 283 (5.8%) |
| Mild TBI | 1,324 (27.1%) | 1,443 (29.5%) |
| Serious TBI | 126 (2.6%) | 130 (2.7) |
| Severe TBI | 77 (1.6%) | 77 (1.6) |
| Facial fracture | 249 (5.1%) | 303 (6.2%) |
| Thoracic injury | 198 (4.1%) | 162 (3.3%) |
| Rib fracture | 541 (11.1%) | 398 (8.1%) |
| Mild abdominal injury | 87 (1.8%) | 89 (1.8%) |
| Severe abdominal injury | 36 (0.7%) | 30 (0.6%) |
| Spinal cord injury | 27 (0.6%) | 10 (0.2%) |
| Stable vertebral fracture or disc injury | 301 (6.2%) | 249 (5.1%) |
| **Injury severity** | 5 (IQR 4–9) | 5 (IQR 2–9) |
| ISS 1–3 | 1,145 (23.4%) | 1,360 (27.8%) |
| ISS 4–8 | 1,597 (32.7%) | 1,320 (27.0%) |
| ISS 9–15 | 1,857 (38.0%) | 1,627 (33.3%) |
| ISS ≥16 | 239 (4.9%) | 194 (4.0%) |
| Missing | 45 (0.9%) | 390 (8.0%) |
| **ICU-admission (yes)** | 358 (7.3%) | 292 (6.0%) |
| Missing | 0 (0.0%) | 0 (0.0%) |

Abbreviations:SES, social-economic status; ICU, intensive care unit; ISS, Injury Severity Score; IQR, Interquartile range; LOS, length of stay; TBI, traumatic brain injury; yrs, years.

Compared to the non-responders, participants were more severely injured and had a higher probability of being admitted to the ICU. In addition, responders had a higher median status score (based on the mean income, % of people with a low income, % of people with low

**Table 2. Mean (SD) summary scores of self-reported health status and psychological outcomes up to 2 years post-trauma.**

| Time post-trauma | EQ-5D-3L* | | HUI2** | | HUI3** | | HADSA*** | | HADSD*** | | IES**** | |
|---|---|---|---|---|---|---|---|---|---|---|---|---|
| | N | Mean (SD) | N | Mean (SD) | N | Mean (SD) | N | Mean (SD) | N | Mean (SD) | N | Mean (SD) |
| 1 week | 1,776 | 0.49 (0.32) | 1,776 | 0.61 (0.22) | 1,776 | 0.38 (0.31) | 1,776 | 4.92 (3.98) | 1,776 | 5.00 (4.28) | 1,776 | 14.80 (15.80) |
| 1 month | 2,971 | 0.56 (0.30) | 2,971 | 0.67 (0.22) | 2,971 | 0.45 (0.35) | 2,644 | 4.81 (3.95) | 2,644 | 4.77 (4.17) | 2,644 | 14.44 (15.73) |
| 3 months | 3,131 | 0.69 (0.27) | 2,735 | 0.72 (0.22) | 2,735 | 0.53 (0.35) | 2,423 | 4.57 (3.80) | 2,423 | 4.24 (4.02) | 2,819 | 12.75 (15.47) |
| 6 months | 3,433 | 0.74 (0.25) | 2,442 | 0.75 (0.22) | 2,442 | 0.58 (0.35) | 2,191 | 4.21 (3.79) | 2,191 | 3.91 (4.01) | 3,182 | 11.42 (15.28) |
| 12 months | 3,151 | 0.76 (0.25) | 2,391 | 0.76 (0.22) | 2,391 | 0.60 (0.36) | 2,165 | 4.32 (3.78) | 2,165 | 3.74 (3.97) | 2,925 | 10.98 (14.98) |
| 24 months | 2,734 | 0.77 (0.26) | 2,137 | 0.77 (0.21) | 2,137 | 0.62 (0.35) | 1,990 | 4.31 (3.76) | 1,990 | 3.62 (3.87) | 2,587 | 10.35 (14.72) |

*Completed by total study population.

**Not administered to patients who completed only the short questionnaire.

***Not administered to proxy participants or to patients who completed only the short questionnaire.

****Not administered to patients aged ≥65 with a hip fracture who completed only the short questionnaire.

Range of EQ-5D-3L: 0–1. Mean EQ-5D-3L of the general Dutch population: 0.87 [47].

Range of HUI 2/3: 0–1

Range of HADSA and HADSD: 0–21

Range of IES: 0–75

Abbreviations: SD, standard deviation; EQ-5D, EuroQol-5D-3L; HUI2/3, Health Utilities Index Mark 2/3; HADS, Hospital Anxiety and Depression Scale; HADSA, Hospital Anxiety and Depression scale, subscale anxiety; HADSD, Hospital Anxiety and Depression scale, subscale depression; IES, Impact of Event Scale.

educational level and % of unemployed people in the neighborhood) compared to the general Dutch population (mean 0.28) and compared to the median status score of the non-responders (median 0.33, min. score -3.03, max. score 2,58). Patients aged 18–44 and ≥85 years showed relatively low response rates (35%-40% and 39%, respectively). Patients with minor injuries (ISS 1–3) revealed a low response rate (46%), as well as patients with a hospital LOS of ≤2 or ≥15 days (46% and 45%, respectively).

## Health status

The mean EQ-5D-3L summary score increased from 0.49 (SD 0.32) at 1 week post-trauma to 0.77 (SD 0.26) at 24 months post-trauma. The mean pre-injury EQ-5D score was 0.85 (SD 0.23). In addition, the mean (SD) HUI2 and HUI3 scores increased from 0.61 (0.22) and 0.38 (0.31) at 1 week post-trauma to 0.77 (0.21) and 0.62 (0.35) at 24 months post-trauma, respectively (see Table 2). With regard to the individual domains of the EQ-5D, trauma patients reported various problems on the 'mobility, 'usual activities' and 'pain/discomfort' dimensions during the 24 months of follow-up (see Fig 2). In addition, during the 24 months, the prevalence of problems on all dimensions of the EQ-5D decreased, but remained higher at 24 months compared to pre-injury (46% and 32%, respectively for mobility, 23% and 16%, respectively for self-care, 44% and 26%, respectively for usual activities, 52% and 32%, respectively for pain/discomfort and 22% and 16%, respectively for anxiety/depression).

## Psychological outcomes

The mean summary scores (SD) of the HADSA (clinical symptoms of anxiety), HADSD (clinical symptoms of depression) and IES (symptoms of post-traumatic stress) slightly decreased from 4.92 (3.98), 5.00 (4.28) and 14.80 (15.80), respectively, at 1 week post-trauma to 4.31 (3.76), 3.62 (3.87) and 10.35 (14.72), respectively, at 24 months post-trauma (see Table 2).

The prevalence rates of clinical symptoms of anxiety (HADSA≥11), depression (HADSD≥11) and post-traumatic stress (IES≥35) reported at 1 week post-trauma were

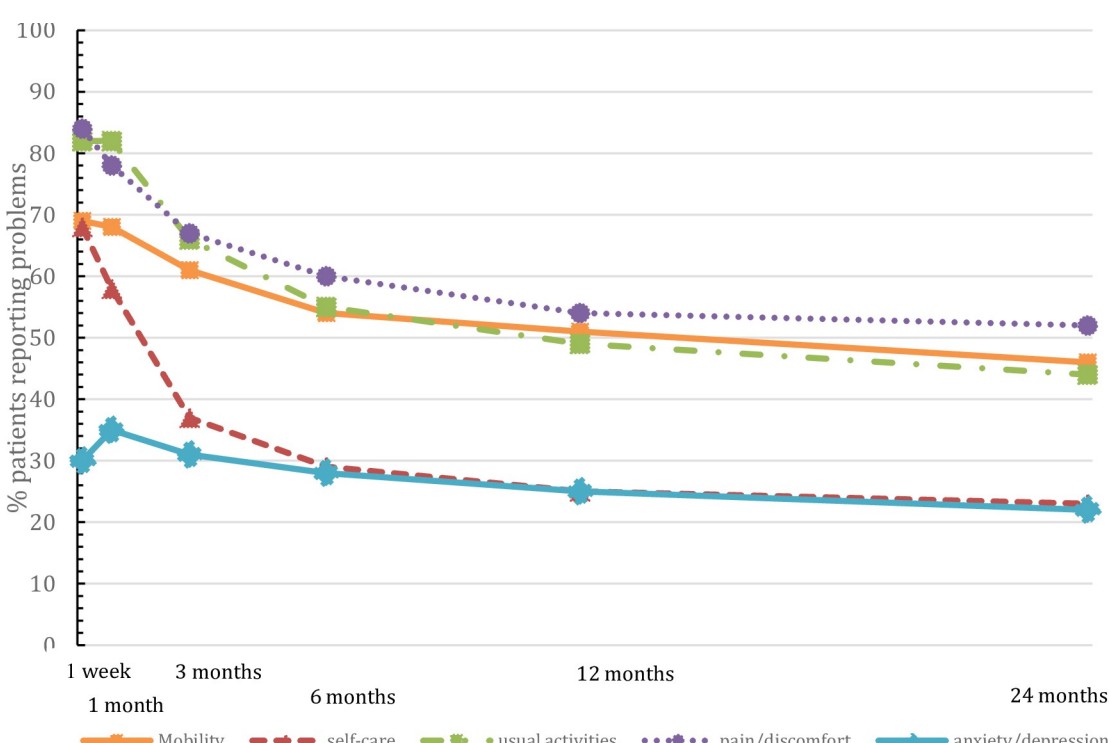

**Fig 2. Prevalence of moderate or severe problems (%) on each EuroQol-5D-3L dimension up until 2 years of follow-up.**

10.2%, 12.3% and 13.5%, respectively, and showed a small decrease over time to 7.8%, 6.8% and 11.0% at 24 months post-trauma, respectively.

The results revealed that patients with symptoms of post-traumatic stress (IES≥35) showed worse outcomes on the EQ-5D-3L than patients with no symptoms of post-traumatic stress (Fig 3).

## Prognostic factors of health status

Overall, HS measured as with the EQ-5D-3L increased over at least up until 6 months for all groups of patients and stabilized between 6 and 12 months post-trauma for most groups (see Table 3). Female patients had a lower HS compared to males at every time point.

At all time points, patients aged 85 and older had the lowest HS compared to the other age categories. At 3 and 6 months, all patient groups between 25 and 74 years reported the same HS whereas patients aged between 18 and 24 reported a higher EQ-5D summary score. HS stabilized at 6 or 12 months for every age group, except for patients between 25 and 44 years for whom HS increased further.

Except for 1 week, patients with a high educational level had the highest HS.

With an increasing number of comorbidities, HS decreased. Patients with moderate injuries (ISS 9–15) showed on almost each time point the lowest HS. At 1 week, severely injured patients (ISS ≥16) showed the lowest mean HS (0.28, SD 0.35).

Patients with the longest hospital LOS (≥15 days) had the lowest mean HS at all time points, ranging from 0.28 (SD 0.34) at 1 week up to 0.62 (SD 0.28) at 24 months after trauma.

After adjustment for confounding factors, short-term (1 week and 1 month) prognostic factors for a significant lower EQ-5D summary score were female gender, a higher number of comorbidities, a longer LOS, a higher ISS, pelvic injury, tibia/complex foot or femur fracture,

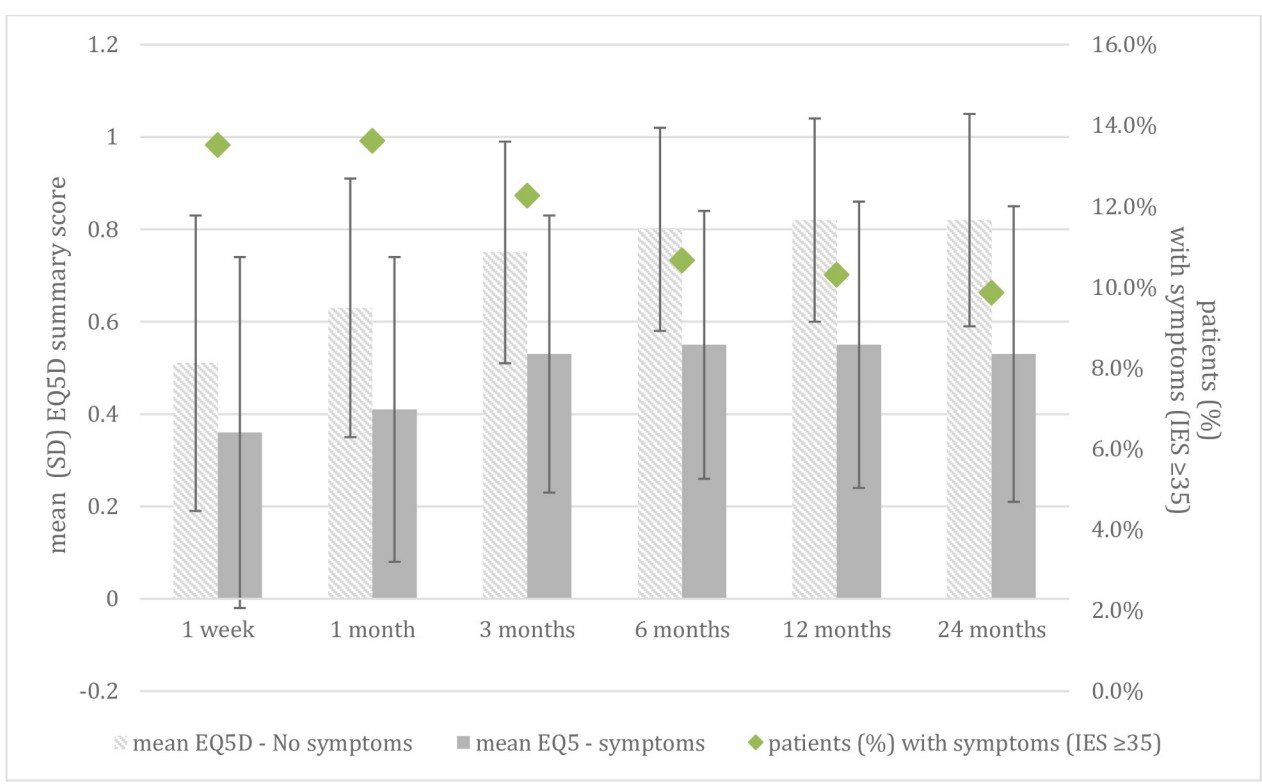

**Fig 3. Mean summary scores of the EuroQol-5D-3L questionnaire and patients reporting symptoms on the Impast of Event Scale.**

radius/ulna/hand fracture, shoulder/upper arm injury, rib fracture, spinal cord injury and stable vertebral fracture/disc injury (see Table 4). Mid-term (3 and 6 months) prognostic factors were a higher number of comorbidities, an ISS between 4 and 15, a longer LOS, radius/ulna/hand fracture, tibia/complex foot or femur fracture, severe TBI, spinal cord injury and stable vertebral fracture/disc injury. Long-term (12 and 24 months) prognostic factors for a lower HS were: age 75 and above, 2 or more comorbidities, a longer LOS, tibia/complex foot or femur fracture, spinal cord injury and stable vertebral fracture/disc injury were prognostic factors. A high educational level was associated with higher HS in the long-term analysis.

## Discussion

This study describes HS and psychological outcomes over 24 months after trauma. HS markedly improved during the 24 months after trauma, most of which occurred within the first 3 months. Compared to pre-injury HS, a large decrease in HS was found 1 week post-trauma. The prevalence rates of symptoms of anxiety and depression were relatively low. In contrast, symptoms of post-traumatic stress were highly prevalent and were present five times as often as compared with the Dutch general population [48]. Additionally, in line with the literature, having symptoms of post-traumatic stress were associated with a lower HS [27]. The addition of an assessment at 1 week post-trauma in the present study adds detailed insight into (baseline) functioning shortly after trauma. Therefore, it provides a more valid assessment of the magnitude of recovery thereafter.

Between 1 week and 3 months, the percentage of patients who reported problems on the 'pain/discomfort' and 'self-care' dimensions of the EQ-5D decreased steeply. For the 'mobility' and 'usual activities' dimensions, this decrease started 1 month after trauma. The percentage

**Table 3. Mean (SD) of the self-reported health status for patient and injury characteristics as measured with the EQ-5D-3L.**

| Characteristics | | Mean (SD) EQ-5D-3L summary score | | | | | | |
|---|---|---|---|---|---|---|---|---|
| | | Pre-injury | 1 week | 1 month | 3 months | 6 months | 12 months | 24 months |
| Gender | Male | 0.90 (0.19) | 0.54 (0.32) | 0.62 (0.28) | 0.74 (0.25) | 0.79 (0.23) | 0.83 (0.21) | 0.84 (0.21) |
| | Female | 0.80 (0.25) | 0.43 (0.31) | 0.50 (0.30) | 0.64 (0.28) | 0.69 (0.26) | 0.70 (0.27) | 0.72 (0.28) |
| Age (yrs) | 18–24 | 0.95 (0,13) | 0.50 (0.32) | 0.63 (0.24) | 0.79 (0.24) | 0.85 (0.21) | 0.86 (0.21) | 0.86 (0.22) |
| | 25–44 | 0.95 (0,12) | 0.45 (0.30) | 0.59 (0.29) | 0.74 (0.24) | 0.79 (0.24) | 0.84 (0.22) | 0.87 (0.19) |
| | 45–64 | 0.91 (0.17) | 0.49 (0.31) | 0.60 (0.27) | 0.73 (0.24) | 0.79 (0.21) | 0.83 (0.21) | 0.83 (0.22) |
| | 65–74 | 0.88 (0.20) | 0.51 (0.31) | 0.62 (0.28) | 0.73 (0.24) | 0.79 (0.22) | 0.80 (0.22) | 0.81 (0.23) |
| | 75–84 | 0.78 (0.26) | 0.52 (0.33) | 0.53 (0.31) | 0.66 (0.27) | 0.70 (0.25) | 0.70 (0.26) | 0.70 (0.28) |
| | ≥85 | 0.63 (0.28) | 0.40 (0.33) | 0.39 (0.32) | 0.50 (0.29) | 0.55 (0.29) | 0.57 (0.29) | 0.56 (0.30) |
| Educational level | Low | 0.78 (0.27) | 0.49 (0.33) | 0.52 (0.31) | 0.65 (0.28) | 0.70 (0.27) | 0.71 (0.27) | 0.72 (0.28) |
| | Middle | 0.90 (0.17) | 0.50 (0.31) | 0.59 (0.29) | 0.71 (0.26) | 0.77 (0.24) | 0.80 (0.23) | 0.81 (0.23) |
| | High | 0.93 (0.13) | 0.48 (0.30) | 0.62 (0.26) | 0.75 (0.23) | 0.80 (0.21) | 0.84 (0.20) | 0.86 (0.20) |
| Comorbidity | 0 | 0.96 (0.12) | 0.53 (0.30) | 0.65 (0.26) | 0.77 (0.23) | 0.83 (0.19) | 0.86 (0.18) | 0.87 (0.18) |
| | 1 | 0.85 (0.21) | 0.49 (0.32) | 0.56 (0.29) | 0.69 (0.25) | 0.75 (0.24) | 0.77 (0.24) | 0.78 (0.25) |
| | 2 | 0.75 (0.24) | 0.45 (0.30) | 0.47 (0.31) | 0.61 (0.29) | 0.66 (0.26) | 0.67 (0.27) | 0.68 (0.28) |
| | ≥3 | 0.64 (0.29) | 0.40 (0.33) | 0.42 (0.32) | 0.56 (0.29) | 0.58 (0.29) | 0.59 (0.29) | 0.59 (0.29) |
| ISS | 1–3 | 0.86 (0.22) | 0.63 (0.29) | 0.69 (0.27) | 0.78 (0.23) | 0.79 (0.24) | 0.79 (0.25) | 0.81 (0.25) |
| | 4–8 | 0.89 (0.20) | 0.46 (0.31) | 0.56 (0.29) | 0.71 (0.25) | 0.77 (0.23) | 0.80 (0.22) | 0.81 (0.23) |
| | 9–15 | 0.80 (0.26) | 0.43 (0.30) | 0.50 (0.30) | 0.63 (0.28) | 0.68 (0.27) | 0.70 (0.28) | 0.72 (0.28) |
| | ≥16 | 0.90 (0.18) | 0.37 (0.35) | 0.50 (0.31) | 0.65 (0.30) | 0.74 (0.25) | 0.77 (0.24) | 0.77 (0.25) |
| LOS (days) | ≤2 | 0.91 (0.18) | 0.61 (0.29) | 0.70 (0.25) | 0.81 (0.21) | 0.83 (0.20) | 0.84 (0.22) | 0.86 (0.21) |
| | 3–7 | 0.85 (0.23) | 0.47 (0.30) | 0.55 (0.28) | 0.69 (0.25) | 0.74 (0.24) | 0.77 (0.24) | 0.79 (0.24) |
| | 8–14 | 0.78 (0.27) | 0.31 (0.31) | 0.46 (0.29) | 0.59 (0.29) | 0.65 (0.27) | 0.68 (0.28) | 0.67 (0.29) |
| | ≥15 | 0.74 (0.28) | 0.28 (0.34) | 0.32 (0.31) | 0.50 (0.29) | 0.58 (0.28) | 0.60 (0.28) | 0.62 (0.28) |

Range of EQ-5D-3L: 0–1. Mean EQ-5D-3L of the general Dutch population: 0.87.

Abbreviations: SES, socio-economic status; SD, standard deviation; EQ-5D, EuroQol-5D-3L; ISS, Injury Severity Score; LOS, length of hospital stay; yrs, years

of patients who reported problems on the 'anxiety and depression' dimension was the highest at 1 month after trauma. Within 6 months post-trauma, patients showed the most recovery. From 6 months post-trauma onwards, little improvement in overall HS was found. The percentage of patients who reported improvements on the different EQ-5D domains increased through the 24 months of follow-up. However, the vast majority of trauma patients did not recover to their pre-injury HS. The mean EQ-5D-3L summary score for the total Dutch adult population is considered 0.87 (SD 0.18) (for males 0.89 (SD 0.16) and for females 0.85 (SD 0.19)) [47].

At short-term (up to 1 month post-trauma) female gender, lower extremity, spine, shoulder and upper arm injuries, injury severity, comorbidities and a longer hospital stay were associated with lower HS. At mid-term (3 and 6 months), almost the same prognostic factors were significant and relevant, however, only injury severity seemed to be less important. In the long-term a greater age, two or more comorbidities, a longer hospital stay and only a few injuries (i.e., lower extremity fracture and spine injury) showed a significantly lower HS. The effect of injury severity seemed to fade over time. Spinal cord injury patients had the highest risk (long-term Beta -0,18, CI -0,27;-0,08) of a lower HS during both the short (not significant), mid and long-term post-trauma. Middle and high educational levels were associated with a higher HS in the long term compared to those with low educational levels.

**Table 4. Multivariable longitudinal analysis of short, mid and long-term prognostic factors for decreased health status as measured with the EQ-5D-3L.**

| | | Number of patients | Total (1 week-24 months) *Beta (95% C.I.) | Short-term (1 week-1 month) *Beta (95% C.I.) | Mid-term (3–6 months) *Beta (95% C.I.) | Long-term (12–24 months) *Beta (95% C.I.) |
|---|---|---|---|---|---|---|
| **Characteristics** | | **4,809** | **n = 4,809** | **n = 3,314** | **n = 4,048** | **n = 3,422** |
| **Gender** | Male | 2,291 | ref | ref | ref | ref |
| | Female | 2,518 | -0.05 (-0.06; -0.04) | -0.06 (-0.08; -0.05) | -0.05 (-0.06;-0.03) | -0.06 (-0.08; -0.05) |
| **Age (yrs)** | 18–44 | 721 | ref | ref | ref | ref |
| | 45–64 | 1,345 | 0.04 (0.02; 0.05) | 0.06 (0.04; 0.09) | 0.02 (-0.00; 0.04) | 0.01 (-0.01; 0.03) |
| | 65–74 | 956 | 0.09 (0.08; 0.11) | 0.14 (0.11; 0.17) | 0.07 (0.05; 0.09) | 0.04 (0.02; 0.06) |
| | ≥75 | 1,787 | 0.02 (0.01; 0.04) | 0.09 (0.06; 0.12) | -0.01 (-0.03; 0.01) | -0.05 (-0.07; -0.02) |
| **Educational level** | Low | 2,637 | ref | ref | ref | ref |
| | Middle | 1,280 | 0.02 (0.00; 0.03) | 0.01 (-0.01; 0.03) | 0.01 (-0.00; 0.02) | 0.03 (0.01; 0.04) |
| | High | 893 | 0.03 (0.02; 0.05) | 0.02 (-0,01; 0,04) | 0.04 (0.02; 0.05) | 0.05 (0.04; 0.07) |
| **Number of comorbidities** | None | 1,801 | ref | ref | ref | ref |
| | 1 | 1,426 | -0.06 (-0.07; -0.05) | -0.06 (-0.08; -0.04) | -0.06 (-0.07; -0.04) | -0.05 (-0.07; -0.04) |
| | ≥2 | 1,582 | -0.15 (-0.16; -0.14) | -0.14 (-0.17; -0.12) | -0.14 (-0.16;-0.13) | -0.16 (-0.18; -0.15) |
| **ISS** | 1–3 | 1,139 | ref | ref | ref | ref |
| | 4–8 | 1,596 | 0.00 (-0.01; 0.02) | -0.03 (-0.06; -0.01) | 0.01 (-0.05; 0.03) | 0.03 (0.01; 0.05) |
| | 9–15 | 1,838 | -0.02 (-0.04; -0.00) | -0.06 (-0.09; -0.02) | -0.02 (-0.04; -0.01) | 0.00 (-0.02; 0.03) |
| | ≥16 | 235 | 0.01 (-0.01; 0.04) | -0.02 (-0.08; 0.04) | 0.01 (-0.04; 0.05) | 0.04 (-0.01; 0.08) |
| **LOS** | ≤2 | 1,425 | ref | ref | ref | ref |
| | 3–7 | 2,053 | -0.05 (-0.06; -0.04) | -0.08 (-0.10; -0.06) | -0.06 (-0.07; -0.04) | -0.03 (-0.04; -0.01) |
| | 8–14 | 975 | -0.11 (-0.12; -0.09) | -0.15 (-0.18; -0.12) | -0.11 (-0.13; -0.09) | -0.08 (-0.11; -0.06) |
| | ≥15 | 357 | -0.20 (-0.22; -0.18) | -0.24 (-0.28; -0.20) | -0.20 (-0.23; -0.17) | -0.17 (-0.20; -0.14) |
| **Injury** | Pelvic injury | 286 | -0.05 (-0.07; -0.03) | -0.13 (-0.16; -0.10) | -0,01 (-0.04; 0.01) | 0.00 (-0.02; 0.03) |
| | Hip fracture | 1,242 | -0.02 (-0.04; -0.01) | -0.02 (-0.05; 0.01) | -0.03 (-0.05; -0.00) | -0.02 (-0.05; 0.00) |
| | Tibia, complex foot or femur fracture | 561 | -0.05 (-0.06; -0.04) | -0.11 (-0.14; -0.08) | -0.05 (-0.07; -0.02) | -0.02 (-0.04; 0.00) |
| | Shoulder and upper arm injury | 469 | -0.03 (-0.04; -0.01) | -0.07 (-0.10; -0.05) | -0.02 (-0.04; 0.00) | -0.00 (-0.02; 0.02) |
| | Radius, ulna or hand fracture | 304 | 0.00 (-0.02; 0.02) | -0.03 (-0.06; 0.00) | 0.00 (-0.02; 0.03) | 0.02 (-0.00; 0.05) |
| | Mild TBI (AIS 1–2) | 1,302 | 0.03 (0.02; 0.04) | 0.06 (0.04; 0.08) | 0.03 (0.02; 0.05) | 0.02 (-0.00; 0.03) |
| | Serious TBI (AIS 3) | 125 | 0.04 (0.01; 0.07) | 0.07 (0.02; 0.13) | 0.05 (0.01; 0.10) | 0.01 (-0.03; 0.05) |
| | Severe TBI (≥ 4) | 76 | -0.03 (-0.06; 0.02) | -0.02 (-0.10; 0.07) | -0.06 (-0.12; 0.00) | -0.01 (-0.07; 0.05) |
| | Facial fracture | 243 | 0.02 (0.01; 0.04) | 0.05 (0.01; 0.09) | 0.02 (-0.01; 0.04) | 0.01 (-0.01; 0.04) |
| | Thoracic injury | 198 | 0.04 (0.02; 0.06) | 0.05 (0.01; 0.10) | 0.04 (0.02; 0.07) | 0.03 (-0.01; 0.06) |
| | Rib fracture | 533 | 0.01 (-0.00; 0.03) | -0.01 (-0.04; 0.01) | 0.03 (0.01; 0.05) | 0.01 (-0.01; 0.04) |
| | Mild abdominal injury | 85 | 0.02 (-0.02; 0.05) | 0.04 (-0.02; 0.10) | 0.01 (-0.04; 0.05) | 0.02 (-0.03; 0.07) |
| | Severe abdominal injury | 36 | 0.03 (-0.02; 0.08) | -0.05 (-0.15; 0.05) | 0.06 (-0.04; 0.13) | 0.05 (-0.02; 0.13) |
| | Spinal cord injury | 27 | -0.11 (-0.17; -0.05) | -0.10 (-0.22; 0.03) | -0.12 (-0.20; -0.03) | -0.15 (-0.23; -0.07) |
| | Stable vertebral fracture or disc injury | 296 | -0.05 (-0.07; -0.03) | -0.08 (-0.12; -0.05) | -0.05 (-0.08; -0.03) | -0.03 (-0.06; -0.01) |

* Mixed models, adjusted for all other variables

Beta: mean increase in EQ-5D-3L score (improvement of Health Status) compared to the reference category.

Range of EQ-5D-3L: 0–1. Mean EQ-5D-3L of the general Dutch population: 0.87 [47].

Abbreviations: ISS, Injury Severity Score; AIS, Abbreviated Injury Scale; LOS, Hospital lenght of stay in days; TBI, traumatic brain injury; yrs = years

Most recovery in HS occurs up to 3 months post-trauma, which is in agreement with previous studies on this topic [9, 10, 19]. In this regard, the addition of an assessment at 1 week post-trauma in the present study adds detailed insight into (baseline) functioning shortly after trauma. Prior work also confirms the finding that a large proportion of patients have a considerably lower HS 1 year post-trauma compared to pre-injury HS [9, 11, 16] or compared to the HS of the general population [49].

In addition to the physical injury itself, other characteristics largely affect HS after trauma. This was particularly in the long-term. This finding extends those of previous studies [15, 35], confirming that patients who were the most severely injured, were not necessarily those with the lowest HS.

In this study, the prevalence of symptoms of anxiety and depression was slightly higher compared to the prevalence of an anxiety disorder or depression in the general Dutch population (both disorders are estimated to be present in 10.0% of the Dutch population) [50, 51]. As a result, the prevalence of symptoms of anxiety and depression slightly decreased over time. Recent and comparable studies have documented a slightly higher prevalence of symptoms of anxiety and depression [9, 16].

The prevalence of symptoms of post-traumatic stress was high as compared with the Dutch population (11.0% at 2 years post-trauma versus 2.6–3.3%) [48]. Our prevalence rate of long-term symptoms of post-traumatic stress is in line with previous research that also uses the ≥35 cut-off point for the IES [13]. However, the systematic review by Haagsma *et al.* [52] revealed prevalence of post-traumatic stress in hospitalized trauma patients that ranged from 30% (90% CI 27%-33%) within 3 months post-trauma to 6% (90% CI 4%-10%) at 1 year. Compared to those results, we found a lower prevalence of symptoms of post-traumatic stress early post-trauma whereas a higher prevalence was found at the 1 year follow-up. This discrepancy may be due to studies using various instruments and different cut-off points to indicate post-traumatic stress.

This study was conducted according to the recommended guidelines for measuring non-fatal outcomes after trauma [41]. The BIOS included a broad study population, measured both short-term and long-term outcomes, measured functioning prior to the trauma, included a large number of patient and injury-related characteristics and requested information from proxy informants for patients who were incapable of completing the set of questionnaires themselves. Recruitment for the BIOS occurred in all hospitals of the Dutch Noord-Brabant region, covering both urban and rural populations.

The inclusion of elderly individuals with a hip fracture in the current study improves the generalizability of the study findings. In this study, one out of four participants was aged 65 or older and had a hip fracture. We are aware that recovery patterns (due to comorbidities and functional decline) in the elderly population are different from recovery patterns in younger patients.

This study also has several limitations. First, there was selection bias since younger patients, elderly patients, patients with very minor injuries (ISS 1–3) and those with a low status score (used as a proxy to indicate SES; a low status score indicate a lower SES-level) were less likely to participate. It is challenging to include these specific groups. Previous studies also reported lower response rates from younger and elderly patients [7, 10, 24, 35], from patients with minor injuries [10] and from those with low educational level [53, 54]. Second, only a selected group of patients (18% of the eligible population) completed the 1 week assessment. Since most recovery occurs within the first 3 months post-trauma, it is vital to examine very early recovery patterns. This study provides unique data since it incorporates the use of a standardized 1 week assessment in a comprehensive group of trauma patients. Nevertheless in most cases, non-responders at this time point felt too disabled to complete a long questionnaire.

Therefore, this most likely led to an underestimations of the reported HS and psychological outcomes 1 week post-trauma. Third, there were many missing data, especially early post-trauma. However, we were able to impute missing sum scores in several cases. Fourth, we did not correct for pre-injury HS in the longitudinal analyses since we were not able to collect pre-injury HS in all participants. Fifth, we did not include a pre-injury assessment of the symptoms of anxiety or depression. Sixth, we cannot exclude change findings due to multiple testing. Change findings might have occurred for the results of the longitudinal analyses as presented in Table 4.

Given the acute nature of trauma, it is difficult to include patients soon after they experience trauma in order to examine very early recovery patterns. To increase the response rate and to reduce loss to follow-up, future research should minimize the large number of questionnaires that patients have to complete at each time point, especially early post-trauma. A promising technique includes computerized adaptive testing. In this already proven valid technique, tailored-made short and precise domain-specific data can be collected [55–58].

We are aware that recall bias and response shift most likely led to an overestimation of the pre-injury HS as measured in this study. However, to produce valid estimates of the health impact and the decrease in functioning after trauma, information on patients functioning prior to the trauma is crucial [59–62]. Future research should focus on the effects of recall bias and response shift on retrospectively collected data.

## Implications for health-care

According to the literature, early recognition, treatment and monitoring of psychological problems improve non-fatal outcomes after trauma [16, 63–66]. Therefore, early screening and interventions to reduce symptoms of post-traumatic stress should be part of standard care. Furthermore in the long long-term, patients aged ≥75 years, patients with a longer length of hospital stay and patients with ≥2 comorbidities are more likely to have a poor HS. For these patients, standard aftercare should be extended to screen for remaining problems that the patients should address after their trauma. For example by a follow-up appointment with a case manager.

Symptoms of post-traumatic stress were frequently reported in this study. Patients need to be better informed about the psychological problems they may experience after trauma. Health-care providers should not solely focus on the physical consequences but should also be aware of all other possible consequences that may occur after trauma. To achieve this, a more holistic approach towards the treatment of trauma patients should be aimed in which the patients' own perspectives on their recovery should play a crucial role.

## Conclusion

Hospitalized trauma patients experience substantial reductions in HS and frequently report symptoms of post-traumatic stress. The results of this study should be interpreted with caution since younger patients, patients with very minor injuries and those with a lower status score (used as a proxy to indicate SES, a lower status score indicates a lower SES-level) were less likely to participate. Most improvements in HS and psychological symptoms occurred within the first 3 months post-trauma. After two years post trauma, the vast majority of trauma patients did not achieve their pre-injury HS. Recovery trajectories varied widely in which female gender, age ≥75 years, spinal cord injury, having more comorbidities, low educational level, a longer hospital stay and symptoms of post-traumatic stress were associated with a higher risk of decreased HS in the long-term after trauma. In the short-term, also several lower extremity injuries are prognostic factors for decreased HS.

## Supporting information

**S1 Table. Injury group classification of the most common types of injury, based on the Abbreviated Injury Score [67].**
(DOCX)

**S2 Table. Missing sum scores of the original data and imputed data of the self-reported health status and psychological measures of the participants of the Brabant Injury Outcome Surveillance (n = 4,883).**
(DOCX)

**S1 File. Methods of imputed data of the Brabant Injury Outcome Surveillance.**
(DOCX)

## Acknowledgments

Memberschip list of the BIOS-group: P.V. van Eerten (Maxima Medical Center, department of surgery, Eindhoven, the Netherlands), F.C. van Eijck (Bravis Hospital, department of surgery, Bergen op Zoom and Roosendaal, the Netherlands), H.J. van Geffen (Jeroen Bosch Hospital, department of surgery, 's-Hertogenbosch, the Netherlands), W.A. Haagh (St. Anna Hospital, department of surgery, Geldrop, the Netherlands), L.M. Poelhekke (Maasziekenhuis Pantein Hospital, department of surgery, Boxmeer, the Netherlands), J.B. Sintenie (Elkerliek Hospital, department of surgery, Helmond, the Netherlands, C.T. Stevens (Bernhoven Hospital, department of surgery, Uden, the Netherlands), A.H. van der Veen (Catharina Hospital, department of surgery, Eindhoven, the Netherlands), C.H. van der Vlies (Maasstad Hospital, department of surgery, Rotterdam, Eindhoven, D.I. Vos (Amphia Hospital, department of surgery, Breda, the Netherlands). Contact person of the BIOS-group: A.H. van der Veen: alexander.vd.veen@-catharinaziekenhuis.nl.

## Author Contributions

**Conceptualization:** Nena Kruithof, Suzanne Polinder, Leonie de Munter, Cornelis L. P. van de Ree, Koen W. W. Lansink, Mariska A. C. de Jongh.

**Data curation:** Nena Kruithof, Suzanne Polinder, Leonie de Munter, Cornelis L. P. van de Ree, Koen W. W. Lansink, Mariska A. C. de Jongh.

**Formal analysis:** Nena Kruithof, Leonie de Munter, Mariska A. C. de Jongh.

**Funding acquisition:** Suzanne Polinder, Koen W. W. Lansink, Mariska A. C. de Jongh.

**Investigation:** Nena Kruithof, Suzanne Polinder, Leonie de Munter, Cornelis L. P. van de Ree, Koen W. W. Lansink, Mariska A. C. de Jongh.

**Methodology:** Nena Kruithof, Suzanne Polinder, Leonie de Munter, Cornelis L. P. van de Ree, Mariska A. C. de Jongh.

**Project administration:** Nena Kruithof, Leonie de Munter, Cornelis L. P. van de Ree, Mariska A. C. de Jongh.

**Resources:** Nena Kruithof, Suzanne Polinder, Leonie de Munter, Cornelis L. P. van de Ree, Mariska A. C. de Jongh.

**Software:** Nena Kruithof, Leonie de Munter, Cornelis L. P. van de Ree, Mariska A. C. de Jongh.

**Supervision:** Suzanne Polinder, Koen W. W. Lansink, Mariska A. C. de Jongh.

**Visualization:** Nena Kruithof, Mariska A. C. de Jongh.

**Writing – original draft:** Nena Kruithof, Suzanne Polinder, Mariska A. C. de Jongh.

**Writing – review & editing:** Nena Kruithof, Suzanne Polinder, Leonie de Munter, Cornelis L. P. van de Ree, Koen W. W. Lansink, Mariska A. C. de Jongh.

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
