## [Decision Letter · Decision Letter 0]

8 Nov 2019

PONE-D-19-21191

Health status and psychological outcome after trauma; a prospective multicenter cohort study

PLOS ONE

Dear MSc. Kruithof,

Thank you for submitting your manuscript to PLOS ONE. After careful consideration, we feel that it has merit but does not fully meet PLOS ONE’s publication criteria as it currently stands. Therefore, we invite you to submit a revised version of the manuscript that addresses the points raised during the review process.

We would appreciate receiving your revised manuscript by Dec 23 2019 11:59PM. To enhance the reproducibility of your results, we recommend that if applicable you deposit your laboratory protocols in protocols.io, where a protocol can be assigned its own identifier (DOI) such that it can be cited independently in the future. For instructions see: http://journals.plos.org/plosone/s/submission-guidelines#loc-laboratory-protocols

We look forward to receiving your revised manuscript.

Kind regards,

Melita J. Giummarra

Academic Editor

PLOS ONE

Journal Requirements:

2. Thank you for including your ethics statement: Ethics committee: Medical Ethics Committee Brabant (project number NL50258.028.14 and NW2016-09).

Prior to participation, participants signed an informed consent form.

Please amend your current ethics statement to confirm that your named institutional review board or ethics committee specifically approved this study.

3. Please carefully proofread your manuscript for typographical errors. For example, on page 7 “a total of 10,227 patients was hospitalized” should be “a total of 10,227 patients were hospitalized”

6. Your ethics statement must appear in the Methods section of your manuscript. If your ethics statement is written in any section besides the Methods, please move it to the Methods section and delete it from any other section. Please also ensure that your ethics statement is included in your manuscript, as the ethics section of your online submission will not be published alongside your manuscript.

7. Please amend your authorship list in your manuscript file to include author Nena Kruithof.

8. One of the noted authors is a group or consortium: BIOS-group. In addition to naming the author group, please list the individual authors and affiliations within this group in the acknowledgments section of your manuscript. Please also indicate clearly a lead author for this group along with a contact email address.

Additional Editor Comments:

In addition to the very helpful comments raised by the reviewers I would like to highlight the following points that require your attention as you revise your manuscript for resubmission.

Methods:

How were comorbidities recorded (e.g., ICD diagnoses?)For the injury groups, it might be better to have a single variable that codes for the broad types of isolated and combined injuries.Please provide a STROBE chart in the body of the paper that includes the exclusions, and the data collection/loss to follow-up at each wave of the study.

Results

In addition to the responder analysis, please include analysis of loss to follow-up. I expect the same factors that differentiate between responders/non-responders will be associated with LTFU.Only summarise data the data that is reported in tables in the text. It is not necessary to fully report the statistics again.Correct the wording in some phrases to reduce ambiguity. For instance in line 212 change “more often” to “higher proportion” rather than  given that the ICU admission was not a frequency WITHIN patients, but between patients. On line 229 change the wording of “most problems” as the data show the proportion of the sample with problems, not what they had the "most problems" with. On line 251 replace “symptoms” with “clinically significant symptoms" given that this is focused on those with symptoms above 11, not any symptoms.Ensure that the terms confounding and covariates are used appropriately.In Table 4 provide columns showing the cell sizes for each level of each factor.

Discussion

There is only very limited discussion of the implications of the research. What new insights does this study add that can be logically and easily included in patient care to improve injury outcomes/reduce injury burden?Note the level of missing data that were imputed was as high as 63.6% for some variables in the limitations

Reviewers' comments:

Reviewer's Responses to Questions

**Comments to the Author**

1. Is the manuscript technically sound, and do the data support the conclusions?

Reviewer #1: Yes

Reviewer #2: Partly

Reviewer #3: Yes

2. Has the statistical analysis been performed appropriately and rigorously? 

Reviewer #1: Yes

Reviewer #2: Yes

Reviewer #3: I Don't Know

3. Have the authors made all data underlying the findings in their manuscript fully available?

Reviewer #1: No

Reviewer #2: Yes

Reviewer #3: No

4. Is the manuscript presented in an intelligible fashion and written in standard English?

Reviewer #1: No

Reviewer #2: No

Reviewer #3: Yes

5. Review Comments to the Author

Reviewer #1: Thank you very much for the opportunity to review this manuscript. The authors present a prospective multi-center cohort study that examined patterns of health status and psychological outcomes (post-traumatic stress, depression, and anxiety) in the 24 months after trauma. Short and long-term prognostic factors for lower health status over time were also investigated. The study adhered to guidelines regarding best practice for the conduct of longitudinal studies examining injury-related disability. Furthermore, the study design ensured that data collection occurred more frequently than many other studies involving trauma populations to date. Investigation of psychological outcomes is a significant strength of the study given the absence of research in this area with respect to general injury.

I believe this manuscript makes an important and novel contribution to the literature. However, there are several issues that could be addressed prior to acceptance for publication.

1) There is significant editing that is required to improve the readability of the manuscript. Perhaps someone independent from the research team could complete a thorough proof-read? There are a number of sentences that need edits in order to make sense. For example, in the abstract, the sentence: “At long-term, higher age, comorbidities, longer hospital stay, lower extremity fracture and spine injury showed lower HS” needs to be modified to read: “At long-term follow-up, participants of higher age, with comorbidities, longer hospital stay, lower extremity fracture and spine injury showed lower HS”. Sentences that begin with ‘besides’ and ‘apparently’ should be removed from the manuscript. The EQ-5D is comprised of five dimensions (as opposed to compressed). There are instances of tense changes in the manuscript also.

2) If possible, the authors should examine the relationship between post-traumatic stress (which was found to be high in the study population) and health status over time. If post-traumatic stress is associated with increased risk of low health status, an important implication is that post-traumatic stress can be addressed early after injury to improve long-term outcome.

3) It would also be good if the authors could justify why they have focused on examining predictors of health status at 24 months but not prognostic factors for anxiety, depression, or post-traumatic stress. It would be interesting to know whether there are different factors that are predictive of these outcomes when compared to health status. This information could be used to identify individuals who are at increased risk of poor psychological outcomes after trauma and to inform intervention development.

3) The authors should provide more of a rationale for including the larger sample of participants who completed only health status assessments (and not the psychological outcome questionnaires) in the analyses for this manuscript, particularly because the focus of the study is on examining psychological outcome after trauma.

4) The reason for some patients (those who did not complete a questionnaire up until 3 months post-trauma) completing a short version of the BIOS-questionnaire is not made explicit. Did this short questionnaire include a pre-injury health status assessment?

5) Please describe how exactly comorbidities were assessed in the questionnaires. Furthermore, what demographic characteristics were collected? Collection of education level is described but other variables examined (such as sex/gender) are not clearly identified.

6) Although the authors describe their multiple imputation process, it is unclear how much missing data there was and how many participant scores needed to be imputed.

7) Tables 2 and 3 could include a pre-injury health status column to aid interpretation.

8) A limitation of the study is the lack of pre-injury assessments of post-traumatic stress, depression, and anxiety.

9) Should pre-injury health status also be examined as a predictor in linear mixed models?

Reviewer #2: I congratulate this research team on collecting data on the long-term impacts of injury. This is not an easy task and provides very valuable information. I have several comments.

1. What does this study add to previous research on the subject? This could be made clearer in the Introduction (justify the need for this study) and the Discussion (what this study adds).

2. Acronyms need to be spelt out in the abstract. Please do not use acronyms that are not well-known to readers. It makes reading the article very arduous.

3. ‘We aimed to describe recovery patterns of health status (HS) and psychological outcomes during 24 months of follow-up and to identify subgroups at risk of both short and long-term health problems after trauma’. Authors could better align the presentation and interpretation of results (abstract and manuscript) to these objectives.

4. ‘Reduction of trauma-related mortality in high-income countries [2] resulted in increased numbers of trauma survivors with long-term injury impact, including reduced health status (HS)’. Can authors provide a reference for the last part of this statement?

5. Document needs extensive grammar/spelling correction

6. Study inclusion criteria are not quite clear to me. Did authors include all presentations to the ED with injury (as stated in the Methods) or just hospital admissions (as stated in the Abstract)? If the latter, did they only include patients with a principal diagnosis of injury? Please clarify.

7. What proportion of the study population were elderly patients with an isolated simple fracture after a fall from their own height? These patients constitute a different population. Please comment.

8. Results – no need to repeat information given in tables. You can just highlight salient results.

9. ‘Compared to the non-responders, participants were more severely injured, were more often admitted to the ICU and had a lower SES’. However, responders had a higher SES score. Please explain how SES status scores should be interpreted.

10. Please provide definitions of mild, serious and severe TBI. These are usually referred to as mild, moderate and severe. Furthermore, as I understand it ‘type of injury’ refers to any injury rather than the most serious or isolated? For example a patient with three injuries including AIS severity score of 5 to the head, another injury of 2 to the head and a tibial fracture would be classed in ‘tibia, complex foot or femur fracture’, ‘mild TBI’ and ‘severe TBI’?

11. In Tables 2 and 3 please add Ns and a note in the legend on how to interpret the scales presented. Means in a healthy population would be useful addition. Why not add pre-injury data here? Also presenting mean differences (inter-patient) with regard to baseline would be interesting. I wonder whether presenting as a Figure would facilitate interpretation (as in Figure 1).

12. It would be useful to have a flow diagram describing numbers included in each phase of the study.

13. Table 4 – Are colours intended to represent statistically significant results? If so, it is applied inconsistently. Furthermore, I would avoid bringing attention to results based on statistical significance only. In addition, given the problem of multiple comparisons, statistical significance of variables with more than two categories should be verified with a global test. Please indicate to readers (in the table legend) how to interpret the betas.

14. Were any sensitivity analyses conducted other than patients with no missing data?

15. Please comment on the external validity of the results given less than 10% of the study population had major trauma.

16. Can authors give any information on the validity of data in the trauma registry?

17. It would be interesting to see results stratified for broad groups of injury: TBI, thoraco-abdo, spinal cord, orthopaedic, multi-system.

18. The summary of key results in line with study objectives (first paragraph of the discussion) may be more effective if it were more succinct.

19. Authors should take account of potential biases (e.g. underestimation of the frequency of health problems due to selection bias) when interpreting results in the Conclusion sections (abstract and manuscript).

Reviewer #3: This manuscript concerns a relevant question on recovery after trauma.

The authors chose a novel approach, intending to include all victims of trauma from a specific region, irrespective of the severity.

The methodology seems to be sound, and the manuscript is fairly easy to read.

I have a number of suggestions to increase reader friendliness, and some comments that I think the authors should address and correct.

First: The inclusion criteria are confusing. You say that “Adults (≥18 years) who visited an emergency department ≤48 hours after trauma were invited to participate” – but all patients included seem to have been admitted, “During the inclusion period of the BIOS, a total of 10,227 patients was hospitalized because of a trauma in one of the participating study centers”. Which group of patients was actually included?

Two: You are using a number of scales/tools (EQ-5D-3L, HUI2/3, HADS-A/D, IES). It would be helpful to readers if you explained a little more detailed how the scores were calculated (EQ-5D-3L – “A summary score of these five dimensions (EQ-5D utility) can be calculated by using the Dutch tariffs [40].”). In addition, I suggest that you inform readers of the “normal values” in the Dutch population of the different scales, as done in the discussion line 336-337 for post-traumatic stress.

Socioeconomic status is not explained. How was the SES assessed, and how did respondents compare to the general population?

The challenge with retrospective assessment of pre-injury health is of course that respondents may see their previous health in light of the present status. This is addressed appropriately in the limitations section.

The tables are numerous and difficult to read when not deeply engaged in the study. I suggest that you elaborate on the legends (e.g. table 4: no explanation of the meaning of the colors). You could also exchange tables for figures. The reader when seeing a figure better perceives a longitudinal change in groups.

In short, an interesting study with need for clarification, and presently looking as you have been working hard and long time with data, and forgotten that potential readers does not have the same confidence with the material.

6. PLOS authors have the option to publish the peer review history of their article (what does this mean?). If published, this will include your full peer review and any attached files.

Reviewer #1: No

Reviewer #2: Yes: Lynne Moore

Reviewer #3: Yes: Torben Wisborg

---

## [Author Response · Author response to Decision Letter 0]

24 Jan 2020

Rebuttal letter

Response to comments

General comments on the manuscript: 

-We placed the DOI-number of the protocol of the Brabant Injury Outcome Surveillance (BIOS) in the manuscript

-Affiliations of the individual members of the BIOS-group are now provided in the acknowledgements. Furthermore, a contact person (including contact details) of the BIOS-group is added

-In this study, there are ethical restrictions on sharing the data set. In the manuscript, we explained in detail were the data can be requested. 

Editor comment:

How were comorbidities recorded (e.g., ICD diagnoses?)

A: In the method section, we now described how self-reported comorbidities were reported (i.e. by using a modified version of the Cumulative Illness Rating Scale). We added this information in the subsection ‘data collection’.

For the injury groups, it might be better to have a single variable that codes for the broad types of isolated and combined injuries.

A; There is an infinite number of combined injuries. Therefore, it is not possible to have a single variable for all the codes. In the analyses, we made corrections for the most common types of injuries and we also made corrections for the Injury Severity Scale. 

Please provide a STROBE chart in the body of the paper that includes the exclusions, and the data collection/loss to follow-up at each wave of the study.

A: In Fig. 1, we now provide a more detailed description of the patient flow of the study. Since patients could flow into the study at 5 different time points, it was not possible to make a STROBE chart. 

In addition to the responder analysis, please include analysis of loss to follow-up. I expect the same factors that differentiate between responders/non-responders will be associated with LTFU.

A: In the second paragraph of the result section, we now provide information about non-responders, responders and those who did not complete the questionnaire at each time point. We described in the second paragraph of the result section that younger patients and those who report to be fully recovered, were most likely to be LTFU. In addition, in the method section, we added the following information: ‘In the BIOS, participants who did not complete a questionnaire were not excluded from the study but they were still invited at the subsequent time points’. 

Only summarise data the data that is reported in tables in the text. It is not necessary to fully report the statistics again.

A: In the result section, we deleted the description of the statistics that were used and we more clearly summarised the most important study findings. 

Correct the wording in some phrases to reduce ambiguity. For instance in line 212 change “more often” to “higher proportion” rather than given that the ICU admission was not a frequency WITHIN patients, but between patients. On line 229 change the wording of “most problems” as the data show the proportion of the sample with problems, not what they had the "most problems" with. On line 251 replace “symptoms” with “clinically significant symptoms" given that this is focused on those with symptoms above 11, not any symptoms.

A: We thank the editor for this specific feedback. We changed the words as suggested above. 

Ensure that the terms confounding and covariates are used appropriately.

A: We have checked this in the manuscript and we did not find inappropriate use of the terms confounding and covariates. 

In Table 4 provide columns showing the cell sizes for each level of each factor.

A: We provided the cell sizes for each level of each factor in Table 4. 

There is only very limited discussion of the implications of the research. What new insights does this study add that can be logically and easily included in patient care to improve injury outcomes/reduce injury burden?

A: We thank the editor for this valuable comment. We extended the discussion with implications for health-care givers.

Note the level of missing data that were imputed was as high as 63.6% for some variables in the limitations.

A: Not all missing values were imputed; only missing sumscores due to missing items were imputed. This resulted in maximal: 13.9% imputed values per variable. We made a mistake by reporting that 63.6% of the data was imputed for some variables. In the discussion, we mentioned the high number of missing data. This was especially the case early post-trauma. 

Reviewer 1: 

There is significant editing that is required to improve the readability of the manuscript. Perhaps someone independent from the research team could complete a thorough proof-read? There are a number of sentences that need edits in order to make sense. For example, in the abstract, the sentence: “At long-term, higher age, comorbidities, longer hospital stay, lower extremity fracture and spine injury showed lower HS” needs to be modified to read: “At long-term follow-up, participants of higher age, with comorbidities, longer hospital stay, lower extremity fracture and spine injury showed lower HS”. Sentences that begin with ‘besides’ and ‘apparently’ should be removed from the manuscript. The EQ-5D is comprised of five dimensions (as opposed to compressed). There are instances of tense changes in the manuscript also.

A: We changed the specific grammatical errors in the sentences as supposed by reviewer. To further improve the readability of the manuscript, correction of grammatical errors and English improvement were made by a native English-speaking person. 

If possible, the authors should examine the relationship between post-traumatic stress (which was found to be high in the study population) and health status over time. If post-traumatic stress is associated with increased risk of low health status, an important implication is that post-traumatic stress can be addressed early after injury to improve long-term outcome.

A: We thank the reviewer for this comment. In the result section, we now described that patients with symptoms of post-traumatic stress reported worse HS compared to patients who did not report symptoms of post-traumatic stress (see also S5 figure for more detailed information). In the conclusion section, we described that symptoms of post-traumatic stress is a determinant of worse HS post-trauma. 

It would also be good if the authors could justify why they have focused on examining predictors of health status at 24 months but not prognostic factors for anxiety, depression, or post-traumatic stress. It would be interesting to know whether there are different factors that are predictive of these outcomes when compared to health status. This information could be used to identify individuals who are at increased risk of poor psychological outcomes after trauma and to inform intervention development.

A:. In the result section, we now described that patients with symptoms of post-traumatic stress reported worse HS compared to patients who did not report symptoms of post-traumatic stress (see also Figure 3 for more detailed information). We also would like to refer the reviewer to a recent BIOS-study publication concerning psychological outcome post-trauma:

-de Munter L, Polinder S, Haagsma J.A, Kruithof N, van de Ree C.L.P, Steyerberg E.W, de Jongh M.A.C. Prevalence and prognostic factors for psychological distress after trauma. Archives of Physical Medicine and Rehabilitation. In Press (23 December 2019).

The authors should provide more of a rationale for including the larger sample of participants who completed only health status assessments (and not the psychological outcome questionnaires) in the analyses for this manuscript, particularly because the focus of the study is on examining psychological outcome after trauma.

A: Patients who completed the short form questionnaire also completed the Impact of Event Scale. Indeed, those patients did not complete the Hospital Anxiety and Depression Scale. Therefore, we created Figure 3 which represents the course of HS of patients with symptoms of post-traumatic stress and the course of HS of patients without symptoms of post-traumatic stress. 

The reason for some patients (those who did not complete a questionnaire up until 3 months post-trauma) completing a short version of the BIOS-questionnaire is not made explicit. Did this short questionnaire include a pre-injury health status assessment?

A: We thank the reviewer for this valuable comment. In the methods section, we now described the reason why patients were invited for the shortened questionnaire; ’Patients who completed the shortened questionnaire included those who stated that the BIOS-questionnaire was to comprehensive to complete or included those who were not traceable by phone and did not returned a BIOS-questionnaire’..We now also described in the methods section that the shortened questionnaire did not include a pre-injury assessment of HS. 

Please describe how exactly comorbidities were assessed in the questionnaires. Furthermore, what demographic characteristics were collected? Collection of education level is described but other variables examined (such as sex/gender) are not clearly identified.

A: We thank the reviewer for this valuable comment. We now described, in the method section, how information regarding comorbidities was collected. Besides, we described in the method section that gender and date of birth were extracted from the self-reported questionnaires. 

Although the authors describe their multiple imputation process, it is unclear how much missing data there was and how many participant scores needed to be imputed.

A: We added an extra appendix (S3 Table) to provide more insights into the missing data and the number and percentage of data that were imputed in this study.

Tables 2 and 3 could include a pre-injury health status column to aid interpretation.

A: In Table 2, we did not provide the pre-injury HS since we only collected the pre-injury HS as measured with the EQ-5D-3L. In Table 3, we now described the pre-injury outcomes. 

A limitation of the study is the lack of pre-injury assessments of post-traumatic stress, depression, and anxiety.

A: We are aware of the added value of a pre-injury assessment of symptoms of post-traumatic stress, depression and anxiety. In the Brabant Injury Outcome Surveillance Study, participants had to complete a comprehensive set of questionnaires (almost 30 pages). Besides, it is not possible to include a pre-injury assessment of post-traumatic stress since we cannot refer to a certain incident that has not taken place yet. In the limitation section we now described: ‘We did not include a pre-injury assessment of symptoms of anxiety or depression.’

Should pre-injury health status also be examined as a predictor in linear mixed models?

A: Pre-injury HS was not examined as a predictor in the linear mixed models. We excluded pre-injury HS in this analysis since we had information regarding pre-injury HS in only 69% of the study population. In the exploratory analyses, we added pre-injury HS as a predictor in the linear mixed models and the variables which explains pre-injury status (such as age and comorbidity) had a small larger effect in the analyses without pre-injury HS. 

Reviewer 2:

What does this study add to previous research on the subject? This could be made clearer in the Introduction (justify the need for this study) and the Discussion (what this study adds).

A: We thank the reviewer for this valuable comment. In the introduction, we described in more detail the specific need for this study. In the discussion, we added some implications for health-care givers, (see the ‘Implications for health care’ paragraph in the discussion section).

Acronyms need to be spelt out in the abstract. Please do not use acronyms that are not well-known to readers. It makes reading the article very arduous.

A: We thank the reviewer for this important and valuable comment. All the acronyms in the abstract are now spelt out. 

‘We aimed to describe recovery patterns of health status (HS) and psychological outcomes during 24 months of follow-up and to identify subgroups at risk of both short and long-term health problems after trauma’. Authors could better align the presentation and interpretation of results (abstract and manuscript) to these objectives.

A: We thank the reviewer for this valuable comment. By providing more information about patients’ psychological functioning post-trauma and its relation to HS (see Fig. 3), we hope that the results better align to the study objective. 

Reduction of trauma-related mortality in high-income countries [2] resulted in increased numbers of trauma survivors with long-term injury impact, including reduced health status (HS)’. Can authors provide a reference for the last part of this statement?

A: In the manuscript, we added two references behind this part of our statement. 

Document needs extensive grammar/spelling correction

A: Correction of grammatical errors and English improvement were made by a native English-speaking person.

Study inclusion criteria are not quite clear to me. Did authors include all presentations to the ED with injury (as stated in the Methods) or just hospital admissions (as stated in the Abstract)? If the latter, did they only include patients with a principal diagnosis of injury? Please clarify.

A: We thank the reviewer for this valuable comment. In the methods, it was unclear that we only included hospitalized patients. We changed this in the method section. 

As stated in the method section: all types of injuries were included, regardless of the intent or severity of the injury. 

What proportion of the study population were elderly patients with an isolated simple fracture after a fall from their own height? These patients constitute a different population. Please comment.

A: One out of four of the included trauma patients in our study was aged 65 or older and had a hip fracture. Indeed, this is a large proportion of the study sample. However, to examine the total burden of trauma patients it is vital to not exclude certain subgroups. 

In the discussion, we added the following text: ‘The inclusion of elderly with a hip fracture in the current study improves the generalizability of the study findings. We are aware that recovery patterns (due to comorbidities and functional decline) in elderly are different from recovery patterns in younger patients.’ 

Results – no need to repeat information given in tables. You can just highlight salient results.

A: In the result section, we deleted a lot of information that was already presented in the tables and figures. 

‘Compared to the non-responders, participants were more severely injured, were more often admitted to the ICU and had a lower SES’. However, responders had a higher SES score. Please explain how SES status scores should be interpreted.

A: We thank the reviewer for this important and valuable comment. Indeed, we made a mistake by reporting that responders had a lower median SES/status score. In the result section, we now changed ‘lower SES’ into ‘higher status score’. We added the following sentence in the result section: ‘non-responders had higher status score compared to the general Dutch population (0.28) and compared to the non-responders’. As we already did in the method section, we now also described in the result section on how to interpret the status score (i.e. Status scores were based on the mean income, % of people with a low income, % of people with low educational level and % of unemployed people in the neighborhood).

Please provide definitions of mild, serious and severe TBI. These are usually referred to as mild, moderate and severe. Furthermore, as I understand it ‘type of injury’ refers to any injury rather than the most serious or isolated? For example a patient with three injuries including AIS severity score of 5 to the head, another injury of 2 to the head and a tibial fracture would be classed in ‘tibia, complex foot or femur fracture’, ‘mild TBI’ and ‘severe TBI’?

A: In the S1 Table, we provided more detail about diagnosis that were considered as mild TBI (AIS severity .1 and .2) or as serious/severe TBI (AIS severity. .3 or higher). Indeed as described in the method section (section prognostic factors), a patient with multiple injuries was classified in one or more injury of the 14 group classifications.

Dit helderder opschrijven in de methods

In Tables 2 and 3 please add Ns and a note in the legend on how to interpret the scales presented. Means in a healthy population would be useful addition. Why not add pre-injury data here? Also presenting mean differences (inter-patient) with regard to baseline would be interesting. I wonder whether presenting as a Figure would facilitate interpretation (as in Figure 1).

A: In Table 2, we did not provide the pre-injury HS since we only collected the pre-injury HS as measured with the EQ-5D-3L, and only for 69% of the study population. In Table 3, we provided the pre-injury scores. In Table 2 and 3, we added the numbers and a note in the legends on how to interpret the scales. 

It would be useful to have a flow diagram describing numbers included in each phase of the study.

A: We included a more precise flow diagram of the study participants in the manuscript (Fig. 1). 

Table 4 – Are colours intended to represent statistically significant results? If so, it is applied inconsistently. Furthermore, I would avoid bringing attention to results based on statistical significance only. In addition, given the problem of multiple comparisons, statistical significance of variables with more than two categories should be verified with a global test. Please indicate to readers (in the table legend) how to interpret the betas.

A: We deleted the colours in Table 4. Indeed due to multiple testing, change findings might have occurred for the results in Table 4. We described the risk of change findings in the limitation section of the discussion. Under Table 4, we shortly described how the Beta should be interpreted (i.e. measures how strong each predictor variable influences the dependent variable.). 

Were any sensitivity analyses conducted other than patients with no missing data?

A: We did not perform any sensitivity analyses. 

Please comment on the external validity of the results given less than 10% of the study population had major trauma.

A: Since we included both rural and urban areas and we included level I, II and III trauma centers, results of this study are at least generalizable to European countries. Nevertheless, we are aware that results may not be generalizable to Australian and USA trauma populations. 

Can authors give any information on the validity of data in the trauma registry?

A; Accurate and consistent registration in the trauma registry is essential. In our study, we used data of the Brabant Trauma Registry, which is part of the Dutch National Trauma Registry (NTR). The registration collaborators of the NTR are well trained before they start to register. Besides, the researchers randomly controlled the data of the trauma registry. In the third paragraph of the method section, we now provide information concerning the validity of the data of the trauma registry. 

It would be interesting to see results stratified for broad groups of injury: TBI, thoraco-abdo, spinal cord, orthopaedic, multi-system.

A: We thank the reviewer for this valuable comment. However, this manuscript contains already a lot of information. It is a good suggestion to stratify broad injury groups. In the future, our research group can examine the effect of stratification on non-fatal outcome after trauma. 

The summary of key results in line with study objectives (first paragraph of the discussion) may be more effective if it were more succinct.

A: We made several changes in the first paragraph of the discussion section in order to provide a short overview of the most important results of our study. 

Authors should take account of potential biases (e.g. underestimation of the frequency of health problems due to selection bias) when interpreting results in the Conclusion sections (abstract and manuscript).

A: We thank the reviewer for this comment. In the conclusion section, we described the high risk of selection bias in our study sample. Due to the restricted word count, we were not able to describe the high risk of selection bias in the abstract. 

Reviewer 3:

The inclusion criteria are confusing. You say that “Adults (≥18 years) who visited an emergency department ≤48 hours after trauma were invited to participate” – but all patients included seem to have been admitted, “During the inclusion period of the BIOS, a total of 10,227 patients was hospitalized because of a trauma in one of the participating study centers”. Which group of patients was actually included?

A: See previous comment of reviewer 2 about the inclusion criteria of participants of the Brabant Injury Outcome Surveillance: In the methods, it was unclear that we only included hospitalized patients. We changed this in the method section. 

You are using a number of scales/tools (EQ-5D-3L, HUI2/3, HADS-A/D, IES). It would be helpful to readers if you explained a little more detailed how the scores were calculated (EQ-5D-3L – “A summary score of these five dimensions (EQ-5D utility) can be calculated by using the Dutch tariffs [40].”). In addition, I suggest that you inform readers of the “normal values” in the Dutch population of the different scales, as done in the discussion line 336-337 for post-traumatic stress.

A: To improve the readability our manuscript, we added extra information about the used questionnaires in the method section. In the discussion, we added information about the Dutch norm scores of the EQ-5D-3L. 

Socioeconomic status is not explained. How was the SES assessed, and how did respondents compare to the general population?

A: In the method section, we added the following sentence: ‘To determine socio-economic status, we used educational level’. Since we had could not collect educational level of the non-responders, we used status scores (based on mean income, % of people with a low income, % of people with low educational level and % of unemployed people in the neighbourhood, see also the first paragraph of the statistical analyses) to indicate SES. In the result section, we added the following sentence: Besides, non-responders had higher statuscores? compared to the general Dutch population (0.28) and compared to the non-responders. 

The challenge with retrospective assessment of pre-injury health is of course that respondents may see their previous health in light of the present status. This is addressed appropriately in the limitations section.

A: We thank the reviewer for this compliment. 

The tables are numerous and difficult to read when not deeply engaged in the study. I suggest that you elaborate on the legends (e.g. table 4: no explanation of the meaning of the colors). You could also exchange tables for figures. The reader when seeing a figure better perceives a longitudinal change in groups.

A: We removed the colours in Table 4. Furthermore, we tried to improve the readability of the tables and figures in our manuscript by adding extra information (e.g. by explaining what a Beta means and by providing the ranges of the questionnaires that were used in the Tables). We did not exchange tables for figures since we think that that may reduce the readability of the findings. 

In short, an interesting study with need for clarification, and presently looking as you have been working hard and long time with data, and forgotten that potential readers does not have the same confidence with the material.

A: Hopefully, we improved the readability of the manuscript by the feedback from the editor and the three reviewers.

---

## [Decision Letter · Decision Letter 1]

10 Feb 2020

PONE-D-19-21191R1

Health status and psychological outcomes after trauma; a prospective multicenter cohort study

PLOS ONE

Dear Dr. Kruithof,

Thank you for submitting your manuscript to PLOS ONE. After careful consideration, we feel that it has merit but does not fully meet PLOS ONE’s publication criteria as it currently stands. Therefore, we invite you to submit a revised version of the manuscript that addresses the points raised during the review process.

We would appreciate receiving your revised manuscript by Mar 26 2020 11:59PM. To enhance the reproducibility of your results, we recommend that if applicable you deposit your laboratory protocols in protocols.io, where a protocol can be assigned its own identifier (DOI) such that it can be cited independently in the future. For instructions see: http://journals.plos.org/plosone/s/submission-guidelines#loc-laboratory-protocols

We look forward to receiving your revised manuscript.

Kind regards,

Melita J. Giummarra

Academic Editor

PLOS ONE

Additional Editor Comments (if provided):

I have a few minor comments that should be addressed, in addition to those raised by the reviewers below, in your revision:

The additional explanation of SES needs to describe whether higher or lower scores indicate greater disadvantage to avoid potential mis-interpretation.The axes on Figure 5 are not clear. Which data do the titles on the left and right-hand x axes correspond to? Moreover, are these truly demonstrating "differences" or mean values? Ideally it is a good idea to depict variability (e.g., sd, se or 95%CI around the mean).Supplementary Figure 5 does not appear to have been uploaded.The explanation of how to interpret a beta score is not sufficient. I suggest you follow the recommendation of Reviewer 2 in providing a clearer explanation of the beta value.

Reviewers' comments:

Reviewer's Responses to Questions

**Comments to the Author**

1. If the authors have adequately addressed your comments raised in a previous round of review and you feel that this manuscript is now acceptable for publication, you may indicate that here to bypass the “Comments to the Author” section, enter your conflict of interest statement in the “Confidential to Editor” section, and submit your "Accept" recommendation.

Reviewer #1: (No Response)

Reviewer #2: (No Response)

Reviewer #3: (No Response)

2. Is the manuscript technically sound, and do the data support the conclusions?

Reviewer #1: Yes

Reviewer #2: Yes

Reviewer #3: Yes

3. Has the statistical analysis been performed appropriately and rigorously? 

Reviewer #1: I Don't Know

Reviewer #2: Yes

Reviewer #3: Yes

4. Have the authors made all data underlying the findings in their manuscript fully available?

Reviewer #1: No

Reviewer #2: Yes

Reviewer #3: No

5. Is the manuscript presented in an intelligible fashion and written in standard English?

Reviewer #1: Yes

Reviewer #2: No

Reviewer #3: Yes

6. Review Comments to the Author

Reviewer #1: I would like to thank the authors for their efforts to improve the manuscript. The limitations associated with the study have now been identified and the implications of the findings are covered in more detail. There is still some language editing required which can be done at the proofing stage if the manuscript is accepted by the journal. One sentence that is important to address in the abstract is: ‘We aimed to describe the recovery patterns of health status (HS) and psychological outcomes during 24 months of follow-up and to identify subgroups at risk of both short and long-term HS after trauma’. The authors should make it clear that the focus is on identification of sub-groups at risk of poor health status.

Reviewer #2: I thank the authors for their thorough responses that address most of my comments. Remaining points are described below.

• The manuscript still requires significant editing for grammatical and spelling errors. For example, in line 71, no comma is required; line 106 should read ‘compiles’ and not ‘complies’; line 140 should read ‘comprise’ and not ‘comproise’. Errors are too extensive to list them all. Spelling/gramma correction software should help (not just the Word tool).

• Line 108. Please explain ‘randomly controlled the data’ in the data verification process more detail. All patient files or a sample? Did you check completeness, consistency, coherence and/or chronology?

• Line 240. Spell out FU. Again, using too many acronyms that are not widely recognized makes reading very arduous

• Table 4 ‘Beta: measures how strong each predictor variable influences the dependent variable’ could be replaced by ‘mean increase in EQ-5D-3L score (improvement in quality of life) compared to the reference category.

Reviewer #3: Thanks for comprehensive responses to the points raised by the reviewers. In general, my concerns have been addressed.

I am still a little confused by the status-scores. Some of the confidence intervals contains negative figures. What are the extremes of the scale? As this seems to be a Dutch scale, you could explain this better to international readers.

In the study design and participants section it is stated: “The Brabant Trauma Registry (BTR) complies pre-hospital and hospital data of all trauma patients admitted after presentation to the ED in the Noord-Brabant region.” You probably intended to say compiles? Same paragraph: “Before the data of the BTR and data of the BIOS-study were merged, the researchers randomly controlled the data of the trauma registry.” This is commendable, but what was “randomly”? 1 of 100, 1 of 1000, 20%? Please elaborate, otherwise this statement is not very descriptive.

7. PLOS authors have the option to publish the peer review history of their article (what does this mean?). If published, this will include your full peer review and any attached files.

Reviewer #1: Yes: Amy Richardson

Reviewer #2: Yes: Lynne Moore

Reviewer #3: Yes: Torben Wisborg

---

## [Author Response · Author response to Decision Letter 1]

20 Mar 2020

Response to comments

Additional Editor Comments

-The additional explanation of SES needs to describe whether higher or lower scores indicate greater disadvantage to avoid potential mis-interpretation.

A: In the ‘statistical analyses’ paragraph, we described the following sentence: a lower status score indicates a lower SES whereas a higher status score indicates a higher SES. Furthermore, we described in the discussion and conclusion paragraphs that a lower status scores is an indicator of a lower SES-level (lines 425 and 473). 

-The axes on Figure 5 are not clear. Which data do the titles on the left and right-hand x axes correspond to? Moreover, are these truly demonstrating "differences" or mean values? Ideally it is a good idea to depict variability (e.g., sd, se or 95%CI around the mean).

A: We changed Figure 3 as supposed by the editor. Furthermore, we made the description of the y-axis more clear. 

-Supplementary Figure 5 does not appear to have been uploaded.

A: In this manuscript, there is no supplementary Figure 5. In the text, we did not refer to Supplementary Figure 5. 

-The explanation of how to interpret a beta score is not sufficient. I suggest you follow the recommendation of Reviewer 2 in providing a clearer explanation of the beta value.

A: See the comment of reviewer 2: we have changed the sentence as supposed by the reviewer. 

Comments reviewer 1

-There is still some language editing required which can be done at the proofing stage if the manuscript is accepted by the journal. 

A: We did made several improvements in language editing throughout the manuscript. 

-One sentence that is important to address in the abstract is: ‘We aimed to describe the recovery patterns of health status (HS) and psychological outcomes during 24 months of follow-up and to identify subgroups at risk of both short and long-term HS after trauma’. The authors should make it clear that the focus is on identification of sub-groups at risk of poor health status.

A: In the abstract and manuscript, we now described: ‘We aimed to 1) identify subgroups at risk of both short and long-term health status (HS) after trauma and 2) to describe the recovery patterns of HS and psychological outcomes during 24 months of follow-up.’

Comments reviewer 2

-The manuscript still requires significant editing for grammatical and spelling errors. For example, in line 71, no comma is required; line 106 should read ‘compiles’ and not ‘complies’; line 140 should read ‘comprise’ and not ‘comproise’. Errors are too extensive to list them all. Spelling/gramma correction software should help (not just the Word tool).

A: We have deleted the comma in line 71, we have changed the word ‘complies’ into ‘compiles’ and changed the word ‘comproise’ into ‘comprise’. Furthermore, we made several improvements in language editing throughout the manuscript. 

-Line 108. Please explain ‘randomly controlled the data’ in the data verification process more detail. All patient files or a sample? Did you check completeness, consistency, coherence and/or chronology?

A: We now described the following sentence: ‘Quality of the data of the BTR and BIOS was checked on outliers and completeness by a trauma coordinator and researcher respectively. Furthermore, data from a sample of the trauma registry was checked manually by a trauma surgeon.’

-Line 240. Spell out FU. Again, using too many acronyms that are not widely recognized makes reading very arduous

A: We have spelled out the FU as ‘follow-up’ as suggested by the reviewer. 

-Table 4 ‘Beta: measures how strong each predictor variable influences the dependent variable’ could be replaced by ‘mean increase in EQ-5D-3L score (improvement in quality of life) compared to the reference category.

A: We thank the reviewer for this suggestion. We changed the sentence as supposed by the reviewer. 

Comments reviewer 3

-I am still a little confused by the status-scores. 

A: In the ‘statistical analyses’ paragraph, we described the following sentence: a lower status score indicates a lower SES whereas a higher status score indicates a higher SES. Hopefully, this will lead to a better interpretation of the status score. Furthermore, we described in the discussion and conclusion paragraphs that a lower status scores is an indicator of a lower SES-level (lines 425 and 473). To interpret the results, we added the minimum and maximum scores.

-Some of the confidence intervals contains negative figures. What are the extremes of the scale? As this seems to be a Dutch scale, you could explain this better to international readers.

A: We added the minimum and maximum score of the EQ-5D-3L under Table 4. 

-In the study design and participants section it is stated: “The Brabant Trauma Registry (BTR) complies pre-hospital and hospital data of all trauma patients admitted after presentation to the ED in the Noord-Brabant region.” You probably intended to say compiles? 

A: Indeed, we intented to say ‘compile’ instead of ‘complies’. We changed the word ‘complies’ into ‘compiles’. 

Same paragraph: “Before the data of the BTR and data of the BIOS-study were merged, the researchers randomly controlled the data of the trauma registry.” This is commendable, but what was “randomly”? 1 of 100, 1 of 1000, 20%? Please elaborate, otherwise this statement is not very descriptive.

A: We deleted the following text: ‘Before the data of the BTR and data of the BIOS-study were merged, the researchers randomly controlled the data of the trauma registry’. We replaced this text with the following sentence: ‘Quality of the data of the BTR and BIOS was checked on outliers and completeness by a trauma coordinator and researcher respectively. Furthermore, data from a sample of the trauma registry was checked manually by a trauma surgeon.’

---

## [Editor Report · Decision Letter 2]

26 Mar 2020

PONE-D-19-21191R2

Health status and psychological outcomes after trauma; a prospective multicenter cohort study

PLOS ONE

Dear Dr. Kruithof,

Thank you for submitting your manuscript to PLOS ONE. After careful consideration, we feel that it has merit but does not fully meet PLOS ONE’s publication criteria as it currently stands. Therefore, we invite you to submit a revised version of the manuscript that addresses the points raised during the review process.

I have uploaded a marked copy of your most recent revision highlighting the relatively minor points that still require further clarification or correction before I can make a decision about the suitability of your submission for publication.

We would appreciate receiving your revised manuscript by May 10 2020 11:59PM. To enhance the reproducibility of your results, we recommend that if applicable you deposit your laboratory protocols in protocols.io, where a protocol can be assigned its own identifier (DOI) such that it can be cited independently in the future. For instructions see: http://journals.plos.org/plosone/s/submission-guidelines#loc-laboratory-protocols

Please include the following items when submitting your revised manuscript:A rebuttal letter that responds to each point raised by the academic editor and reviewer(s). This letter should be uploaded as separate file and labeled 'Response to Reviewers'.A marked-up copy of your manuscript that highlights changes made to the original version. This file should be uploaded as separate file and labeled 'Revised Manuscript with Track Changes'.An unmarked version of your revised paper without tracked changes. This file should be uploaded as separate file and labeled 'Manuscript'.

We look forward to receiving your revised manuscript.

Kind regards,

Melita J. Giummarra

Academic Editor

PLOS ONE

---

## [Author Response · Author response to Decision Letter 2]

28 Mar 2020

Response to comments

-In Fig. 3, we changed the word ‘’sumscore’’ into ‘’summary score’’, as suggested by the editor.

-In the abstract and in the introduction, we changed the description of the aim of the study, as suggested by the editor. 

-In line 111, we added the following information on how data was checked: ‘Data of the BTR (i.e. Brabant Trauma Registry) was checked as part of routine practice while the data of the BIOS was checked randomly by the researchers.’

-In line 140, we removed the question mark as supposed by the editor. 

-In one of the previous versions of our submitted manuscript, we referred to S5. This was not correct, we should have referred to Fig 3 instead of S5. In the current manuscript, we now correctly refer to Fig 3.

-In line 398, we removed the ‘’s’’ as supposed by the editor. 

-In line 457, we removed the ‘’n’’ as supposed by the editor.

---

## [Editor Report · Decision Letter 3]

30 Mar 2020

Health status and psychological outcomes after trauma; a prospective multicenter cohort study

PONE-D-19-21191R3

Dear Dr. Kruithof,

We are pleased to inform you that your manuscript has been judged scientifically suitable for publication and will be formally accepted for publication once it complies with all outstanding technical requirements.

With kind regards,

Melita J. Giummarra

Academic Editor

PLOS ONE
---

## [Editor Report · Acceptance letter]

2 Apr 2020

PONE-D-19-21191R3 

Health status and psychological outcomes after trauma; a prospective multicenter cohort study 

Dear Dr. Kruithof:

I am pleased to inform you that your manuscript has been deemed suitable for publication in PLOS ONE. Congratulations! Your manuscript is now with our production department. 

With kind regards,

on behalf of

Dr. Melita J. Giummarra 

Academic Editor

PLOS ONE